# Exploring the Therapeutic Potential of the DOT1L Inhibitor EPZ004777 Using Bioinformatics and Molecular Docking Approaches in Acute Myeloid Leukemia

**DOI:** 10.3390/cimb47030173

**Published:** 2025-03-04

**Authors:** Mehmet Kivrak, Ihsan Nalkiran, Hatice Sevim Nalkiran

**Affiliations:** 1Department of Biostatistics and Medical Informatics, Faculty of Medicine, Recep Tayyip Erdogan University, 53020 Rize, Türkiye; 2Department of Medical Biology, Faculty of Medicine, Recep Tayyip Erdogan University, 53020 Rize, Türkiye; ihsan.nalkiran@erdogan.edu.tr (I.N.); hatice.sevim@erdogan.edu.tr (H.S.N.)

**Keywords:** acute myeloid leukemia, transcriptomics, bioinformatics, molecular docking

## Abstract

Background: Acute myeloid leukemia (AML) is a malignancy characterized by the clonal expansion of hematopoietic stem and progenitor cells, often associated with mutations such as NPM1. DOT1L inhibitors have shown potential as new therapeutic opportunities for NPM1-mutant AML. The aim of this study was to investigate potential alternative targets of the small-molecule inhibitor EPZ004777, in addition to its primary target, DOT1L, using RNA sequencing data from the NCBI-GEO database (GSE85107). Methods: Differentially expressed genes (DEGs) were identified through bioinformatic analysis, followed by pathway enrichment analysis to uncover the relevant biological pathways. Additionally, molecular docking analysis was conducted to assess the binding affinity of EPZ004777 with the proteins CT45A3, HOXA4, SNX19, TPBG, and ZNF185, which were identified as significantly DEGs. The protein structures were obtained from AlphaFold and the Protein Data Bank. Results: EPZ004777 significantly altered gene expression. Oncofetal genes (CT45A3, TPBG) and genes associated with oncogenic pathways (HOXA4, ZNF185, SNX19) were downregulated, while the pro-apoptotic gene BEX3 was upregulated. Pathway enrichment analysis revealed the suppression of the Rap1 signaling pathway and cell adhesion molecules, which may reduce the invasiveness of AML cells. Additionally, upregulation of immune-related pathways suggests enhanced anti-tumor immune responses. Molecular docking analysis demonstrated that EPZ004777 has strong binding potential with SNX19, TPBG, and ZNF185 proteins. Conclusions: EPZ004777 has been identified as a potent modulator of SNX19, TPBG, and ZNF185 associated with apoptosis and tumor progression in AML.

## 1. Background

Acute myeloid leukemia (AML) is characterized by clonal proliferation of hematopoietic stem cells and progenitor cells. Genomic abnormalities confer a selective growth advantage to these cells, inhibiting normal hematopoiesis. AML accounts for approximately 80% of acute leukemia cases in adults [1]. Currently, AML is defined by various cytogenetic and molecular abnormalities, which play an important role in guiding diagnostic approaches and treatment plans due to their prognostic and therapeutic implications [2].

Nucleophosmin (NPM) is predominantly located in the nucleus, though it regularly transports between the nucleus and the cytoplasm [3,4]. This protein is essential for various cellular functions, such as centrosome duplication, maintaining genomic stability, and responding to nucleolar stress [5]. NPM is one of the most commonly mutated genes, occurring in about 30% of adult leukemia cases [3]. NPM1 mutations (NPM1mut) also lead to the cytoplasmic delocalization of NPM1mut [6]. Aberrant cytoplasmic mislocalization of NPM1mut disrupts cellular processes, leading to imbalances such as uncontrolled centrosome proliferation and repression of tumor suppressor genes [6].

NPM1mut directly contributes to the elevated expression of homeobox (HOX) genes, which are essential for sustaining the undifferentiated state of leukemia cells [7]. NPM1mut is also associated with the high expression of HOX gene cofactors MEIS1 and PBX3 [8]. The exact role of NPM1 mutations in the development of AML is not yet completely understood, making it challenging to create targeted treatment strategies [9]. Promising molecules for targeting NPM1mut in AML therapy include ATRA/ATO [10], EAPB0503 [11], NSC348884 [12], EPZ004777, MI-2-2, and MI-503 [13]. EPZ004777 has been identified as a potent inhibitor of DOT1L and has been shown to inhibit leukemogenic gene expression, particularly in cells carrying Mixed Lineage Leukemia translocations, by blocking H3K79 methylation in in vivo studies [14]. DOT1L inhibitors have shown potential as new therapeutic opportunities for NPM1-mutant AML, both as monotherapy and in combination therapies [13].

In this study, the effects of the DOT1L inhibitor EPZ004777 on the NPM1-mutant AML cell lines AML2 and AML3 were evaluated by analyzing RNA sequencing data obtained from the NCBI-Gene Expression Omnibus (GEO) database titled ‘Targeting Chromatin Regulators: Inhibition of Leukemogenic Gene Expression in NPM1 Mutant Leukemia’ (accession code: GSE85107). The resulting gene expression profiles were examined to contribute to the understanding of the molecular mechanisms associated with AML.

## 2. Methods

### 2.1. Dataset and RNA-Seq Data Analysis

In this study, RNA sequencing data (accession code: GSE85107), publicly available from the NCBI-GEO database under the title ‘Targeting Chromatin Regulators: Inhibition of Leukemogenic Gene Expression in NPM1 Mutant Leukemia,’ were utilized to compare the OCI-AML2 and OCI-AML3 human AML cell lines, both derived from *Homo sapiens* [13]. The experimental design of the GSE85107 dataset included treating cells with 10 µM EPZ004777 or DMSO (carrier control) for 7 days in triplicates, resulting in 12 samples (Table 1). High-throughput sequencing was performed to generate gene expression profiles. ERCC synthetic spike-in controls were incorporated into the cell lysis buffer for each sample to normalize cell counts and RNA extraction, as described in the methodology of the original study. Untreated OCI-AML2 replicates were collected for RNA purification to detect fusions [13].

Initial data processing identified 35,237 genes across 12 samples. Dimensionality reduction via Principal Component Analysis (PCA) narrowed this to 16,050 genes for downstream analysis. Among these, 14,476 genes were mapped to Ensembl gene identifiers, while the remaining 1574 genes were retained using their original identifiers. The processed data matrix containing normalized gene expression levels was utilized for exploratory analysis, differential gene expression (DEG) identification, and pathway enrichment analysis using R (version: 3.20) and Bioconductor (version: 4.4.0) packages. This analytical approach ensures robust hypothesis generation and provides insights into the key gene expression changes associated with EPZ004777 treatment. A flowchart summarizing the workflow is illustrated in Figure 1.

### 2.2. Data Preprocessing

During the preprocessing phase, the gene identifiers were matched with a database for automatic conversion and species identification. Genes with minimal downregulation were excluded, and the data were transformed as needed using various methods. When skewed distributions were detected, logarithmic transformation was applied to facilitate analyses that assume approximate normality. This transformation reduced variability while preserving small values and had a significant impact on the downregulated genes. Raw read counts reflected the number of sequencing reads generated under different experimental conditions, indicating that technical variations might arise from sample preparation or sequencing processes (Figure 2A). The distribution of gene expression levels following normalization and transformation (Figure 2B) showed consistent distributions across samples, demonstrating effective minimization of technical variability. The density plot of the transformed gene expression data (Figure 2C) displayed the distribution of expression levels across the dataset, with slight deviations from normality in some samples indicating variability in expression patterns. Additionally, the classification of genes based on their functional categories (Figure 2D) provided insights into the relative abundance of protein-coding genes, pseudogenes, long non-coding RNAs, and microRNAs, offering a comprehensive overview of genomic composition. These methods ensured reliable data preprocessing and normalization, providing a robust foundation for downstream analyses.

### 2.3. PCA

In this study, Principal Component Analysis (PCA) was used to simplify the dataset by deriving principal components (PCs) as linear combinations of the original variables [15]. PCA was conducted on the gene expression matrix derived from the RNA sequencing data to explore the effects of the DMSO and EPZ004777 treatments on the AML2 and AML3 cell lines. The PCA score plot in Figure 3A illustrates the separation of the samples based on treatment and cell line. Clear clustering along the PC1 and PC2 axes demonstrates distinct gene expression profiles between AML2 and AML3. PC1, which explains 68.8% of the total variance, primarily distinguishes between the two cell lines, while PC2, accounting for 16.04% of the variance, contributes to further separation within each treatment group. The clustering of samples treated with DMSO and EPZ004777 within each cell line indicates that treatment differences are less evident in terms of variance captured by the first two principal components. Figure 3B shows that the first two principal components capture the majority of the variance in the dataset, with PC1 and PC2 explaining 84.8% of the total variation. The remaining principal components contribute minimal variance, as indicated by the sharp decline in the explained variation after PC2. These results indicate that PCA effectively distinguishes gene expression profiles between the AML cell lines, with the primary driver of variation being the differences between AML2 and AML3, while treatment with EPZ004777 versus DMSO has a smaller impact on the variance captured by PC1 and PC2.

### 2.4. Molecular Docking Analysis

Molecular docking analysis was conducted to explore the binding interactions between EPZ004777 (PubChem ID: 56962336) (Figure 4) and multiple target proteins, including TPBG (PDB ID: https://doi.org/10.2210/pdb4CNM/pdb) and DOT1L (PDB ID: https://doi.org/10.2210/pdb4ER3/pdb). The three-dimensional structures of the other proteins were not available in any experimental database; therefore, their structural predictions were based solely on AlphaFold models. The models provided by AlphaFold were used as the basis for the relevant analyses. The three-dimensional structures of CT45A3 (AlphaFold ID: AF-Q8NHU0-F1-v4), BEX3 (AlphaFold ID: AF-Q00994-F1-v4), HOXA4 (AlphaFold ID: AF-Q00056-F1-v4), SNX19 (AlphaFold ID: AF-Q92543-F1-v4), ZNF185 (AlphaFold ID: AF-O15231-F1-v4) were obtained from the AlphaFold Protein Structure Database (https://alphafold.ebi.ac.uk/) (accessed on 12 December 2024) and utilized for further analysis.

The ligand structure was retrieved from the PubChem database in SDF format [16]. To prepare the ligand for docking, it was first converted to PDB format using OpenBabel (version 2.4.1) [17] and subsequently subjected to energy minimization using Chimera (version 1.18) [18]. The minimized structure was then converted to PDBQT format using AutoDock Tools 1.5.7 [19]. This step ensured that the ligand was in its most stable conformation prior to docking. The 3D structures of the target proteins were retrieved from AlphaFold [20] and the Protein Data Bank [21] and prepared by removing water molecules, adding polar hydrogens, merging non-polar hydrogens, and assigning Kollman charges using AutoDock Tools 1.5.7. Docking studies were carried out using AutoDock Vina (version 1.1.2) to predict the optimal binding conformation of the ligand within the respective binding sites. A grid spacing of 0.375 Å was applied for all docking calculations, ensuring sufficient resolution to capture precise binding interactions [19]. Separate grid boxes were configured for each protein with the parameters detailed in Table 2, including the center coordinates (X, Y, Z) and dimensions (Å) defined for molecular docking simulations.

Docking calculations for each protein were performed under default exhaustiveness or set to 8 using AutoDock Vina [19]. The docking results were ranked based on binding energy values, and the lowest-energy conformations for each protein–ligand complex were selected for further analysis. Visualization of docking poses, protein–ligand interactions, and hydrogen bond distances were analyzed using BIOVIA Discovery Studio 2024 Client (Dassault Systèmes, 2024), following standard criteria. Key interactions, including hydrogen bonds, hydrophobic contacts, and π-based interactions, were identified and compared across the proteins to understand the differential binding mechanisms of EPZ004777. This docking study provides a comparative analysis of the binding interactions of EPZ004777 with multiple target proteins, offering insights into its potential as a broad-spectrum or selective inhibitor/modulator for CT45A3, HOXA4, SNX19, TPBG, and ZNF185.

## 3. Results

### 3.1. Correlation Matrix Between Cell Lines

The gene expression profiles of the AML2 and AML3 cell lines treated with either DMSO or EPZ004777 were analyzed using log2-transformed values and scatter plots to assess linear relationships (Figure 5). Pearson correlation coefficients (R) and *p*-values were calculated for each replicate to quantify the strength of these relationships. In Figure 5A,C,E, the DMSO-treated groups showed strong positive correlations across all the replicates (R = 0.84–0.89, *p* < 0.01), indicating highly similar and reproducible gene expression patterns between AML2 and AML3 under control conditions. For the EPZ004777-treated groups (Figure 5B,D,F), the correlations were slightly lower (R = 0.82–0.85, *p* < 0.01) but remained robust, reflecting consistent transcriptional responses to the treatment across the biological replicates. The slightly reduced correlation in the EPZ004777-treated samples compared to the DMSO-treated samples suggests that EPZ004777 induces more variability in gene expression, potentially reflecting its regulatory effects on transcriptional mechanisms in AML cells. Despite this variability, the high Pearson correlation coefficients across the replicates highlight the reproducibility of the experimental design and the coherent impact of EPZ004777 on gene expression in AML2 and AML3 cell lines.

### 3.2. DEG2 (Differentially Expressed Genes2) Analysis

The analysis of gene expression differences between the DMSO- or EPZ004777-treated samples identified a substantial number of DEGs. A total of 3578 genes were identified as differentially expressed, with 1725 genes being upregulated and 1853 downregulated (Figure 6A). Figure 6B illustrates the distribution of these genes based on their log2 fold change and adjusted *p*-value. Upregulated genes exhibit significant positive fold changes, while downregulated genes exhibit significant negative fold changes. There are also genes present that did not show significant differential expression (gray). A number of genes exhibit large fold changes, with some having log2 fold changes greater than 20, indicating a substantial difference in gene expression between the DMSO- or EPZ004777-treated groups. The findings shown in Figure 6B demonstrate that a majority of the upregulated and downregulated genes have both statistically significant *p*-values and a notable fold change, further supporting the effect of EPZ004777 treatment on gene expression.

### 3.3. DEG2 Profiles in AML2 and AML3 Cell Lines Following EPZ004777 Treatment

The heatmap in Figure 7A and the DEG analysis results in Table 3 reveal significant differences in gene expression between the AML2 and AML3 cell lines treated with DMSO and EPZ004777. Hierarchical clustering (Figure 7A) shows distinct patterns of gene expression, particularly in AML3 under EPZ004777 treatment. The provided bar charts (Figure 7B–D) illustrate the normalized gene expression levels in the AML2 and AML3 cell lines treated with DMSO or EPZ004777. Overall, EPZ004777 treatment caused significant alterations in gene expression, particularly in AML3, highlighting its differential effects on the two cell lines. In Figure 7B, genes such as BCL9, GSPT2, HOXB9, TMEM215, WASF3, and WFS1 exhibited significantly higher expression in AML3 compared to AML2 under EPZ004777 treatment (*p* < 0.001). Similarly, in Figure 7C, genes such as MGMT, MTMR8, PPFIA2, STAC, TCEAL9, and TMEM54 displayed significantly higher expression in AML3 compared to AML2 under EPZ004777 treatment, with notable increases in expression levels (*p* < 0.001 for most genes). In Figure 7D, genes such as HOXA4, CT45A3, SNX19, TPBG, and ZNF185 were highly expressed in both cell lines under DMSO but showed significant downregulation under EPZ004777 treatment in AML3 (*p* < 0.001). Conversely, the genes BEX3, PARD3, and TUB exhibited upregulated expression levels, and EPZ004777 significantly increased the expression of these genes in AML3 (*p* < 0.001). The diverse responses of these genes to EPZ004777 treatment suggest a cell line-specific modulation of gene expression, particularly in AML3, where EPZ004777 had a more considerable effect on genes involved in various oncogenic pathways. These findings indicate that EPZ004777 may modulate the expression of key regulatory genes in a cell line-dependent manner, with AML3 showing greater sensitivity to the effects of EPZ004777 on gene expression, which could have therapeutic implications for targeting specific genetic pathways in AML.

### 3.4. Pathway Enrichment Analysis Reveals Differential Regulation of Signaling and Immune Pathways in AML Cell Lines Following EPZ004777 Treatment

Pathway enrichment analysis of the DEGs between AML2 and AML3 cell lines treated with DMSO and EPZ004777 revealed several key pathways that were significantly downregulated or upregulated (Figure 8, Table 4). Figure 8A,B shows the enriched pathways associated with downregulated genes, while Figure 8C,D highlights the pathways enriched for upregulated genes. In the downregulated pathways (Figure 8A,B), the ‘Rap1 signaling pathway’ was among the most significantly impacted, with a fold enrichment of 2.279, 43 genes involved, and an −log10(FDR) of 4.8 Other important downregulated pathways included ‘Cell adhesion molecules’ (fold enrichment = 2.112, 27 genes, −log10(FDR) = 1.97) and ‘Neuroactive ligand-receptor interaction’ (fold enrichment = 2.245, 35 genes, −log10(FDR) = 3.61). These pathways, which are involved in cell communication, adhesion, and signaling, showed reduced activity in AML3 compared to AML2 following EPZ004777 treatment. Figure 8B ranks these pathways by −log10(FDR), with the ‘Rap1 signaling pathway’ and ‘Pathways in cancer’ showing the highest fold enrichment and statistical significance. For the upregulated genes (Figure 8C,D), several immune-related pathways were significantly enriched. ‘Autoimmune thyroid disease’ demonstrated the highest fold enrichment of 5.631, involving 14 genes with an −log10(FDR) of 12.8, followed by ‘Allograft rejection’ (fold enrichment = 5.406, 14 genes, −log10(FDR) = 11.9) and ‘Proteoglycans in cancer’ (fold enrichment = 5.019, 28 genes, −log10(FDR) = 1.75). These findings suggest that EPZ004777 treatment may enhance immune-related processes in AML3. Figure 8D shows the fold enrichment of these pathways, with ‘Proteoglycans in cancer’ and ‘Hematopoietic cell lineage’ pathways demonstrating significant changes in expression levels, reflected in their high fold enrichment values (Table 4). Table 4 summarizes the top downregulated and upregulated pathways, presenting their fold enrichment, number of associated genes, and statistical significance. The ‘Rap1 signaling pathway’ and ‘Cell adhesion molecules’ were the most significantly downregulated, with ‘Autoimmune thyroid disease’ and ‘Allograft rejection’ showing the most significant upregulation. These findings indicate that EPZ004777 treatment results in the inhibition of essential signaling pathways related to cancer progression and cell adhesion, while activating immune-related pathways that may contribute to the therapeutic effects of the treatment.

### 3.5. Network Analysis

Protein–protein interaction (PPI) network analysis was performed to explore the relationships among DEGs in the AML2 and AML3 cell lines following EPZ004777 treatment. The PPI networks represent the interactions between significantly altered proteins (Figure 9). In Figure 9A, the network shows interactions between proteins encoded by genes such as BCL9, WASF3, PPFIA2, GSPT2, and TUB, forming a tightly connected network that suggests strong functional interdependencies. BCL9 stands out as a central node, indicating its potential importance as a regulatory protein in this network. Figure 9B expands the network to include additional proteins, including HOXA4, TMEM215, HOXB9, and MTMR8, resulting in a more intricate interaction map. TMEM215 and HOXA4 emerge as central hubs connecting several other proteins, suggesting that EPZ004777 treatment may affect a wide array of biological processes by modulating these protein interactions. These findings highlight significant protein interaction clusters, particularly centered around BCL9, TMEM215, and HOXA4, which may play key roles in mediating the response of AML cells to EPZ004777 treatment. The strong connectivity observed among these proteins suggests that important pathways and mechanisms are being affected by the treatment, warranting further investigation into their biological functions and potential as therapeutic targets.

### 3.6. Molecular Docking of EPZ004777 with Target Proteins

To identify the potential target proteins of EPZ004777, a differential gene expression analysis was performed using RNA sequencing data. Genes that were significantly downregulated following EPZ004777 treatment in AML cells were prioritized. Based on their biological relevance, oncofetal genes and genes involved in tumor progression were selected. Structural information for the selected proteins was obtained from reliable databases such as the Protein Data Bank (PDB) and AlphaFold; AlphaFold models were used when structural data were limited. Based on these criteria, the identified proteins were evaluated as potential targets of EPZ004777 and selected for further molecular docking analyses. In the molecular docking analysis, the interactions of EPZ004777 with target proteins were examined, and binding energy, the number of hydrogen bonds, and interaction distances were evaluated (Table 5). According to the results, the DOT1L protein demonstrated the strongest binding energy at −9.7 kcal/mol. Five hydrogen bonds were formed between DOT1L and EPZ004777, with binding distances ranging from 2.28 Å to 3.63 Å. This finding confirms the effectiveness of EPZ004777 as a DOT1L inhibitor and highlights the importance of DOT1L as a target protein. The SNX19 protein demonstrated strong binding with a binding energy of −8.2 kcal/mol and three hydrogen bonds, indicating a stable interaction with the ligand. Similarly, the TPBG protein exhibited a notable interaction with a binding energy of −7.7 kcal/mol. By forming seven hydrogen bonds, TPBG stood out as the protein with the highest number of hydrogen bonds in the analysis. These results suggest that both SNX19 and TPBG proteins have significant potential as alternative targets for EPZ004777 beyond DOT1L. The ZNF185 protein showed moderate binding, with a binding energy of −7.2 kcal/mol and the formation of six hydrogen bonds. These results suggest that ZNF185 may serve as a stable interaction site for the ligand. On the other hand, the CT45A3 (−6.7 kcal/mol) and HOXA4 (−6.5 kcal/mol) proteins exhibited lower binding energies and formed two hydrogen bonds each, indicating limited interaction with the ligand. The BEX3 protein displayed the weakest binding energy at −5.2 kcal/mol. However, the formation of four hydrogen bonds suggests that the interaction is not entirely weak but limited in terms of stability (Table 5). In conclusion, this analysis confirmed the DOT1L protein as the strongest target for EPZ004777. TPBG and SNX19 proteins also demonstrated significant binding capacities. The use of DOT1L as a control in this study validated the inhibitory effect of EPZ004777 and enabled the evaluation of potential binding regions in other target proteins.

As a known DOT1L inhibitor, EPZ004777 was included in this study as a reference to validate the molecular docking analyses conducted with other target proteins. EPZ004777 was stably positioned within the DOT1L binding pocket, forming key interactions with the surrounding amino acid residues. Figure 10A illustrates the overall docking pose of EPZ004777 within the DOT1L binding site, showing the ligand appropriately positioned in the pocket. Figure 10B provides a close-up view of the binding pocket, highlighting specific interactions between EPZ004777 and key residues in DOT1L. Figure 10C presents a 2D interaction diagram detailing the binding interactions. Conventional hydrogen bonds were observed between EPZ004777 and GLU186 and ASP222, contributing significantly to binding stability. This docking analysis highlights the high binding affinity of EPZ004777 for DOT1L and provides insights into the molecular interactions that support this strong ligand–protein association.

The molecular docking interactions between EPZ004447 and the SNX19 protein are detailed in Figure 11. The overall positioning of EPZ004447 within the SNX19 binding site is shown in Figure 11A, demonstrating that the ligand fits well into the binding pocket and forms a stable interaction network with surrounding amino acids. Figure 11B provides a close-up view of the binding site, highlighting additional key interactions. Figure 11C presents a two-dimensional interaction diagram, illustrating the detailed binding interactions between EPZ004447 and SNX19. Conventional hydrogen bonds between EPZ004447 and ASN A:245 and PRO A:891 are prominently displayed, significantly contributing to the stability of the ligand–protein complex.

The molecular docking interactions between EPZ004447 and the TPBG protein are presented in Figure 12, emphasizing the key binding characteristics. Figure 12A illustrates the overall docking pose of EPZ004447 within the TPBG binding site, showing the ligand appropriately positioned in the pocket. Figure 12B provides a close-up view of the binding pocket, highlighting specific interactions between EPZ004447 and key residues in TPBG. Figure 12C presents a 2D interaction diagram detailing the binding interactions, including conventional hydrogen bonds and other stabilizing forces. EPZ004447 forms conventional hydrogen bonds with ASN A:192, HIS A:221, ASN A:220, ARG A:193, GLN A:191, ASP A:188, and GLU A:196, which significantly contributes to the stability of the ligand within the pocket. This analysis highlights the strong and stable interaction network between EPZ004447 and TPBG, driven by a combination of hydrogen bonds and other stabilizing interactions, showing the specificity of the ligand for TPBG.

The molecular docking interactions between EPZ004447 and the ZNF185 protein are detailed in Figure 13, highlighting the key interactions. Figure 13A illustrates the overall docking pose of EPZ004447 within the ZNF185 binding site, showing the ligand fitting well into the pocket and forming a stable interaction network with the surrounding amino acid residues. Figure 13B provides a close-up view of the binding pocket, highlighting specific interactions between EPZ004447 and key residues in ZNF185. Figure 13C presents a 2D interaction diagram, detailing the binding interactions and the stabilizing network formed within the pocket. EPZ004447 forms conventional hydrogen bonds with TYR A:581, ILE A:674, HIS A:646, and TYR A:685, which significantly contribute to the stability of the ligand–protein complex. This analysis emphasizes the strong and stable interaction network between EPZ004447 and ZNF185, driven by a combination of hydrogen bonds and other stabilizing interactions. The critical residues within the binding pocket play a key role in ensuring high-affinity and specific binding of the ligand, underscoring the robustness of this molecular docking result.

Detailed insights into the molecular docking interactions between EPZ004777 and the CT45A3 protein are presented in Appendix A, highlighting the key binding characteristics. Appendix A illustrates the overall docking pose of EPZ004777 within the binding site of CT45A3, showing the ligand well-positioned on the protein surface and interacting with critical residues within the active site. Appendix A provides a close-up view of the binding site, highlighting specific interactions between EPZ004777 and key residues in CT45A3. Appendix A presents a 2D interaction diagram detailing the binding interactions. EPZ004777 forms conventional hydrogen bonds with HIS A:180, which significantly contribute to the stability of the ligand within the binding pocket. The key interactions between EPZ004447 and the HOXA4 protein are detailed in Appendix A, emphasizing the structural basis of the binding. Appendix A illustrates the overall docking pose of EPZ004447 within the HOXA4 binding pocket, demonstrating that the ligand fits well into the site. Appendix A provides a close-up view of the binding site, highlighting specific interactions between EPZ004447 and key residues in HOXA4. Appendix A presents a 2D interaction diagram detailing the binding interactions. EPZ004447 forms conventional hydrogen bonds with LYS A:199 and LYS A:200, which play a central role in stabilizing the ligand within the binding pocket. The binding dynamics of EPZ004777 with the BEX3 protein are outlined in Appendix A, focusing on key molecular interactions. Appendix A illustrates the overall binding pose of EPZ004777 within the BEX3 protein, showing the ligand specifically occupying the active site and interacting with critical residues. Appendix A provides a close-up view of the binding pocket, highlighting specific interactions between EPZ004777 and key residues in BEX3. Appendix A presents a 2D interaction diagram detailing the binding interactions. EPZ004777 forms conventional hydrogen bonds with ASN A:46 and PHE A:47, which are central to stabilizing the ligand within the binding pocket.

Across all the proteins analyzed, additional interactions such as Pi–Pi stacking, Pi–Alkyl, and hydrophobic contacts contribute significantly to the overall stability and specificity of the ligand–protein complexes. While the specific residues involved in these interactions vary, their role in enhancing the binding stability and ensuring proper positioning of the ligands within the binding pockets is critical, underscoring the robust nature of these molecular docking results.

## 4. Discussion

AML, a diverse form of hematologic cancer, is the most prevalent type of acute leukemia in adults [22]. NPM1 is involved in several biological processes, such as mRNA splicing, chromatin remodeling, embryogenesis, tumor suppression, and cell apoptosis [23]. NPM1 mutations are a specific finding for AML [3]. However, the exact mechanism of action of NPM1 mutations in AML is not well understood, and further exploration of therapeutic strategies is still needed [24]. The histone methyltransferase activity of DOT1L is crucial for the correct regulation of the HOXA gene cluster in hematopoietic cells, and inhibitors of DOT1L decrease the expression of both HOXA and MEIS genes [13]. EPZ004777 is one of the potential molecules for targeting NPM1mut in AML [13]. In this study, we reanalyzed RNA-seq data (GSE85107) to evaluate the effects of EPZ004777 on AML cells using bioinformatics and molecular docking analyses. Our findings confirmed the role of DOT1L pathways in regulating HOX and FLT3 expression, consistent with the reference study. However, we also identified alternative molecular targets of EPZ004777 beyond DOT1L, including SNX19, TPBG, and ZNF185, which exhibited strong binding interactions in molecular docking analyses. Additionally, we observed that EPZ004777 downregulated oncogenes such as HOXA4, TPBG, SNX19, and ZNF185, while upregulating the pro-apoptotic gene BEX3. Notably, the inhibition of the Rap1 signaling pathway was associated with reduced cell invasion and enhanced immune response, suggesting a potential immunomodulatory effect. These findings provide new insights into the mechanism of action of EPZ004777 and highlight additional therapeutic targets in NPM1-mutated AML.

In the study conducted by Daigle and colleagues, the therapeutic potential, mechanism of action, and applicability of EPZ004777 in cancer treatment were comprehensively investigated. The study demonstrated that EPZ004777 is a potent inhibitor of DOT1L, targeting the S-adenosylmethionine (SAM) binding site with high specificity. Additionally, EPZ004777 was found to exhibit significant selectivity for DOT1L compared to other lysine methyltransferases. This high specificity is clinically significant, as EPZ004777 has been shown to reduce H3K79 methylation levels, selectively inhibit the proliferation of MLL-rearranged leukemia cells, and induce notable transcriptional deregulation. Furthermore, EPZ004777 has been reported to inhibit the expression of HOXA9 and MEIS1 genes in a concentration-dependent manner. Studies conducted in cell culture and mouse xenograft disease models have demonstrated that DOT1L inhibition via EPZ004777 provides a strong molecular basis for the development of targeted therapeutic approaches against MLL-rearranged leukemias. However, the poor pharmacokinetic properties of EPZ004777 limit its clinical application. Consequently, research efforts continue to focus on the development of new DOT1L inhibitors with higher potency and improved drug-like properties. [14]. In another study, EPZ004777-mediated DOT1L inhibition was found to be effective in increasing E-cadherin levels in breast cancer cells by suppressing key transcription factors involved in the epithelial–mesenchymal transition (EMT) [25]. Additionally, EPZ004777 treatment has been observed to provide therapeutic benefits in various solid malignancies where DOT1L plays a significant role. One example of this is colorectal cancer (CRC), where the application of EPZ004777 to multiple CRC cell lines led to a significant reduction in cell viability and tumorigenicity, and these findings were further validated in in vivo xenograft models [26]. Furthermore, EPZ004777 has been reported to indirectly demonstrate that the oncogenic activity of DOT1L is associated with androgen receptor (AR) status. Studies have shown that EPZ004777 significantly suppresses colony formation and cell viability in AR-positive prostate cancer (PCa) cells compared to AR-negative cells. These effects have also been validated through in vivo applications in xenograft mouse models [27]. EPZ004777 is a potential therapeutic molecule for NPM1-mut AML cells, and, in this study, the effects of EPZ004777 on OCI-AML2 and OCI-AML3 cell lines were examined.

### 4.1. Effects of EPZ004777 on Gene Expression

Bioinformatics analyses show that EPZ004777 treatment leads to the overexpression of the BEX3 gene [28] enhancing apoptosis processes in cancer cells. The overexpression of BEX3 highlights the potential of EPZ004777 to inhibit cancer cell survival pathways and promote apoptosis. Additionally, the expression of tumor-associated genes such as CT45A3 [29], TPBG [30], HOXA4 [8,31], ZNF185 [32], and SNX19 [33] was significantly downregulated. The treatment with EPZ004777 induces the overexpression of the BEX3 gene in cancer cells, thereby promoting apoptosis and inhibiting survival pathways in cancer cells. This finding supports the role of BEX3 as a pro-apoptotic gene and suggests that tumor formation is suppressed in breast cancer mouse models [28,34]. High expression of CT45A3 has been associated with disease progression and poor prognosis in ovarian cancer [29]. The decrease in the expression of CT45A3 following EPZ004777 treatment could suggest that the tumor progression could be impeded by the treatment.

TPBG is associated with various cellular processes, including cell migration, morphological changes, and membrane integrity [35]. TPBG expression has been reported to induce vascular remodeling in the tumor microenvironment, thereby playing a crucial role in tumor growth and invasion [36]. TPBG has been identified as an oncofetal antigen that is highly expressed in various types of cancer [37,38]. For instance, high TPBG expression has been observed in tumor tissues, including bladder, breast, ovarian, pancreatic, and gastric carcinomas, and it has been closely associated with poor clinical outcomes in colorectal, ovarian, and gastric cancers [38,39,40,41]. Furthermore, TPBG, a glycoprotein that inhibits the Wnt/β-catenin pathway and is associated with poor clinical outcomes in various cancer types [30,39,40,41], shows decreased expression following EPZ004777 treatment, which may contribute to the inhibition of cancer cell proliferation.

Sorting nexins (SNXs) are a protein family that facilitates intracellular trafficking and signaling [42,43]. However, there is limited data in the literature regarding the relationship between SNX19 and cancer. A study in the literature reported that SNX19, along with 18 other genes, was significantly overexpressed in thyroid oncocytic adenomas compared to the control group [44]. In another study, EPZ004777 treatment significantly downregulated SNX19, which was previously shown to be overexpressed by microarray expression analysis in AML [45]. This suggests that EPZ004777 may exert a regulatory effect on SNX19 expression, potentially contributing to the inhibition of cancer cell proliferation and survival in acute myeloid leukemia.

Studies on the biological and clinical significance of ZNF185 are limited. In the existing literature, studies investigating the relationship between ZNF185 and the clinical characteristics of cancer have reported that ZNF185 plays different roles depending on the type of cancer. For instance, some studies have suggested that ZNF185 acts as a tumor suppressor in prostate cancer [46], while others have proposed that the low expression of ZNF185 serves as a negative prognostic marker for colorectal cancer [32]. Knockdown of ZNF185 may promote chemosensitivity, apoptosis, and proliferation inhibition [47]. Additionally, in cases of prostate cancer, cranio-cervical squamous cell carcinoma, and non-small-cell lung cancer, ZNF185 expression has been shown to be significantly lower (downregulated) in healthy (non-cancerous) tissues [46,48,49,50]. Finally, the overexpression of HOXA4 promotes cell proliferation and stem cell self-renewal in tumors such as glioma and colorectal cancer [8,31,51,52]. These findings suggest that EPZ004777 has the potential to inhibit cancer cell survival by modulating these genes, which may be a promising approach for therapeutic strategies.

### 4.2. Pathway Enrichment Implications

Pathway enrichment analyses revealed that EPZ004777 treatment significantly altered key signaling pathways and immune-related pathways. Specifically, the downregulation of pathways such as ‘Rap1 signaling’ and ‘Cell adhesion molecules’ suggests that EPZ004777 may disrupt communication and invasiveness in AML cells. Moreover, the upregulation of immune-related pathways indicates that EPZ004777 could enhance anti-tumor immune responses. The potential for EPZ004777 to increase immune responses represents an important finding, particularly in the context of immunotherapeutic approaches to AML. The ‘autoimmune thyroid disease’ pathway may be linked to acute myeloid leukemia (AML), as treatment with EPZ004777 has been shown to activate immune-related pathways in AML cells. This suggests that EPZ004777 could induce immune modulation mechanisms within the AML microenvironment. EPZ004777 alters the epigenetic regulation of AML cells, specifically through DOT1L inhibition. It is known that epigenetic changes can affect not only AML-specific but also immune system-related gene expression regulations. This is particularly important for the regulation of inflammatory responses and immune tolerance mechanisms. For example, the allograft rejection pathway is specifically associated with the T cell-mediated immune response. The altered immune recognition mechanisms of AML may alter interactions between immune cells and leukemic cells. Graft-versus-host disease (GVHD) is a serious immunological complication that occurs when donor immune cells attack recipient tissues. Allo-HSCT is a common method used in the treatment of AML. Cytokines, T cell signaling pathways, and the inflammatory responses involved in the GVHD pathway seem to be related to the immunological mechanisms affected by EPZ004777 treatment. It shows that it changes the immune environment of AML and may increase its interaction with immune cells. This finding suggests that EPZ004777 may not only directly suppress AML cells but also cause immunological changes in the leukemic microenvironment. Therefore, the alteration of the GVHD pathway and related immunological processes may be important for understanding the connection between epigenetic regulation and immune activation in the treatment of AML.

### 4.3. Insights from Molecular Docking Analysis

Molecular docking analysis revealed that EPZ004777 forms interactions with target proteins that exhibit varying degrees of stability and specificity. SNX19 demonstrated the strongest interaction, with a binding energy of −8.2 kcal/mol and three hydrogen bonds, establishing a well-formed and stable interaction network within its binding site. TPBG showed significant binding affinity, with a binding energy of −7.7 kcal/mol and seven hydrogen bonds, highlighting its stability and specificity as a target due to the high number of hydrogen bonds. ZNF185, with a binding energy of −7.2 kcal/mol and six hydrogen bonds, exhibited moderate stability, suggesting its potential as a secondary target for EPZ004777. Proteins with lower binding affinities, such as CT45A3 (−6.7 kcal/mol) and HOXA4 (−6.5 kcal/mol), formed two hydrogen bonds each, while BEX3 (−5.2 kcal/mol) displayed four hydrogen bonds, indicating interactions supported by additional forces such as hydrophobic contacts and Pi stacking. These findings highlight SNX19 as the strongest interacting target protein, with TPBG and ZNF185 also emerging as promising candidates. Although CT45A3 and BEX3 showed more limited interactions, they still possess notable stability potential. These results emphasize the multi-target potential of EPZ004777 and underscore the need for further investigation into the functional roles of SNX19, TPBG, and ZNF185.

### 4.4. Clinical Relevance and Future Research Directions

EPZ004447, on the other hand, demonstrated meaningful interactions with multiple proteins such as SNX19, TPBG, and ZNF185, highlighting its potential to affect several targets. Targets like CT45A3 and HOXA4 displayed moderate interaction profiles but indicated therapeutic potential due to their roles in critical cellular processes. Despite its weaker interaction profile, the involvement of BEX3 in essential pathways suggests it could contribute to broader combination strategies in AML treatment. These findings suggest that EPZ004777 could be utilized as a therapeutic strategy in NPM1-mutant AML, particularly by modulating gene expression profiles and influencing survival and tumor progression pathways. Investigating the roles of other target proteins, such as SNX19, TPBG, ZNF185, HOXA4, CT45A3, and BEX3, could open new avenues for the development of innovative treatments. In conclusion, EPZ004777 emerges as a promising therapeutic agent for AML by targeting DOT1L to modulate key cellular pathways. Exploring the roles of additional targets and integrating them into combination therapies could significantly enhance the efficacy of AML treatments. Leveraging strategies such as epigenetic regulation, cancer stem cell targeting, and precision drug delivery systems will be crucial in optimizing these approaches for clinical applications.

In this study, the potential interactions of EPZ004777 with TPBG, ZNF185, and SNX19 were evaluated using computational approaches. Molecular docking analyses revealed that EPZ004777 has significant binding affinities with these proteins; however, experimental validation is required to confirm these interactions and their biological relevance. Future studies should include Western blot, qPCR, and functional cell-based assays to investigate the effects of EPZ004777 on the expression and activity of these target proteins. Molecular docking is a powerful tool for predicting ligand–protein interactions; however, it has certain limitations, such as assuming a rigid protein structure and not accounting for the dynamic nature of biological systems. Molecular dynamics (MD) simulations could provide further insights into the stability of EPZ004777–protein interactions over time. Our pathway analysis suggested that EPZ004777 downregulates the Rap1 signaling pathway and cell adhesion molecules, potentially reducing AML cell invasiveness. However, whether EPZ004777 interacts directly or indirectly with TPBG, ZNF185, and SNX19 remains unclear. Further studies utilizing ChIP-seq or RNA-seq could clarify the molecular mechanisms underlying these interactions. Additionally, site-directed mutagenesis and functional assays are required to confirm the biological significance of the binding sites of EPZ004777 on TPBG, ZNF185, and SNX19. Determining the functional effects of these binding sites will help elucidate the potential regulatory roles of EPZ004777 on these proteins.

Furthermore, additional experimental validation and preclinical investigations are necessary to assess the feasibility of considering these proteins as therapeutic targets in human cancers. Understanding their roles in different malignancies and their potential as biomarkers or druggable targets would provide valuable insights into their clinical relevance. In conclusion, this study is the first to computationally predict the potential interactions of EPZ004777 with TPBG, ZNF185, and SNX19, identifying them as promising therapeutic targets. However, in vitro and in vivo validation is required to confirm these findings. Future studies will focus on experimental validation, molecular dynamics simulations, and mechanistic analyses to further investigate the therapeutic potential of EPZ004777 in AML.

## 5. Conclusions

In this study, we demonstrated that EPZ004777 exhibits strong molecular docking interactions with TPBG, ZNF185, and SNX19, identifying these proteins as potential key targets. EPZ004777 downregulated tumor-associated genes such as TPBG, ZNF185, and SNX19, suggesting its ability to inhibit tumor progression. The high binding affinities observed with these proteins, coupled with their stable interaction profiles, highlight their relevance as potential biomarkers and therapeutic targets. Additionally, the modulation of these targets indicates the potential of EPZ004777 to disrupt critical pathways involved in tumor growth and survival. Overall, these findings show the promise of EPZ004777 as a therapeutic agent, with TPBG, ZNF185, and SNX19 serving as focal points for further research to refine its application in cancer treatment.

## Figures and Tables

**Figure 1 cimb-47-00173-f001:**
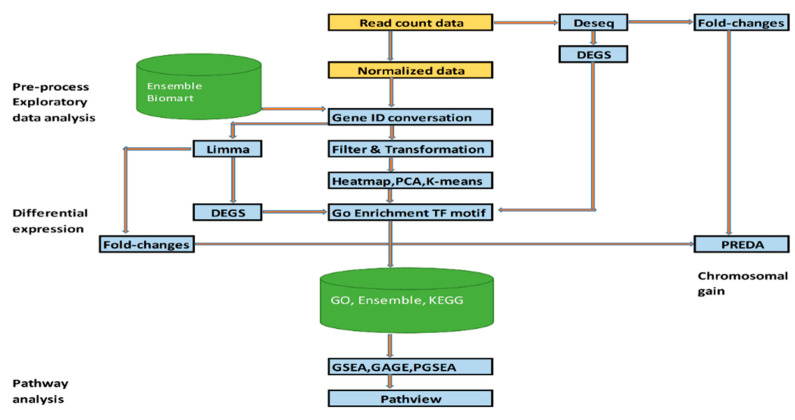
Workflow diagram for normalized expression and RNA-seq read counts.

**Figure 2 cimb-47-00173-f002:**
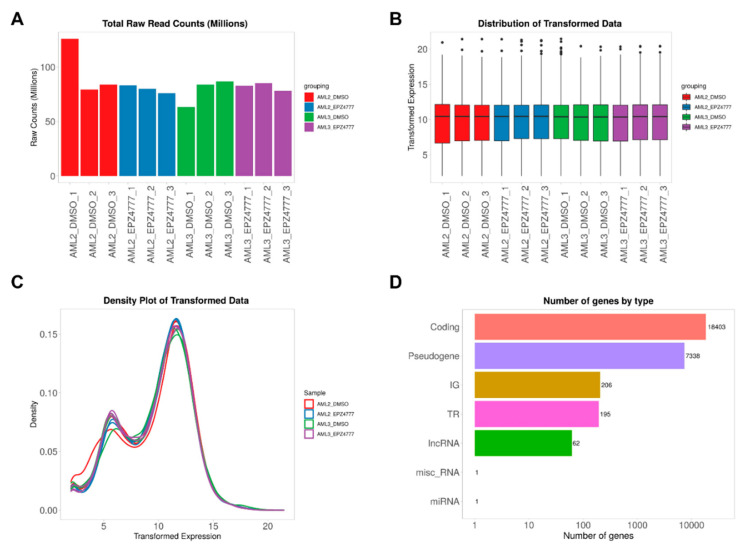
Overview of RNA sequencing data and gene expression distributions in the AML2 and AML3 cell lines treated with DMSO or EPZ004777. (**A**) Total raw read counts (millions) across the AML2 and AML3 samples, showing consistent sequencing depth across the treatments. (**B**) Box plot of log2-transformed gene expression, illustrating a similar distribution across the samples, confirming successful normalization. (**C**) Density plot of the transformed data, indicating consistent gene expression distributions across all the samples. (**D**) Bar graph showing the number of genes by type, with protein-coding genes being the most abundant, followed by pseudogenes and other gene types.

**Figure 3 cimb-47-00173-f003:**
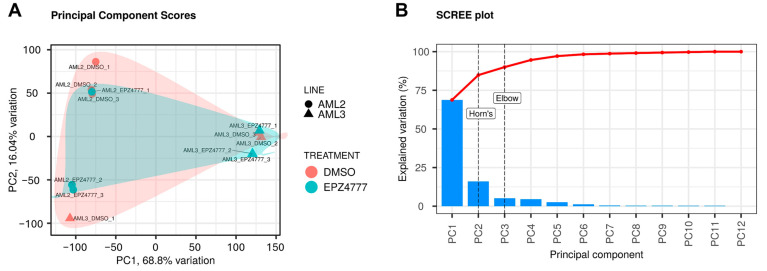
Principal component analysis and investigation of treatment effects on the AML cell lines using PCA. PCA analysis was performed on the gene expression matrix obtained from the RNA-seq data. (**A**) The PCA score plot shows the effects of the DMSO and EPZ004777 treatments on the AML2 and AML3 cell lines. Samples from different treatments and cell lines cluster in distinct regions along the PC1 and PC2 axes. (**B**) presents a scree plot, which indicates that the first two principal components explain a significant portion of the variance in the data.

**Figure 4 cimb-47-00173-f004:**
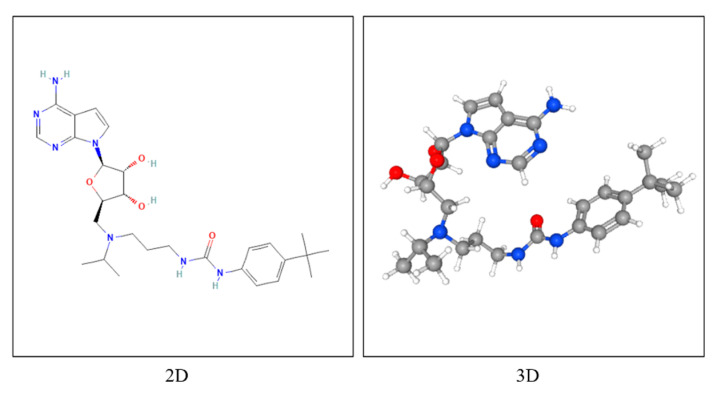
Two- and three-dimensional chemical structure representations of EPZ004777, a chemical compound with PubChem CID: 56962336. The left panel illustrates the 2D structure, displaying the molecular connectivity, while the right panel shows the 3D configuration, highlighting the spatial arrangement of atoms within the molecule.

**Figure 5 cimb-47-00173-f005:**
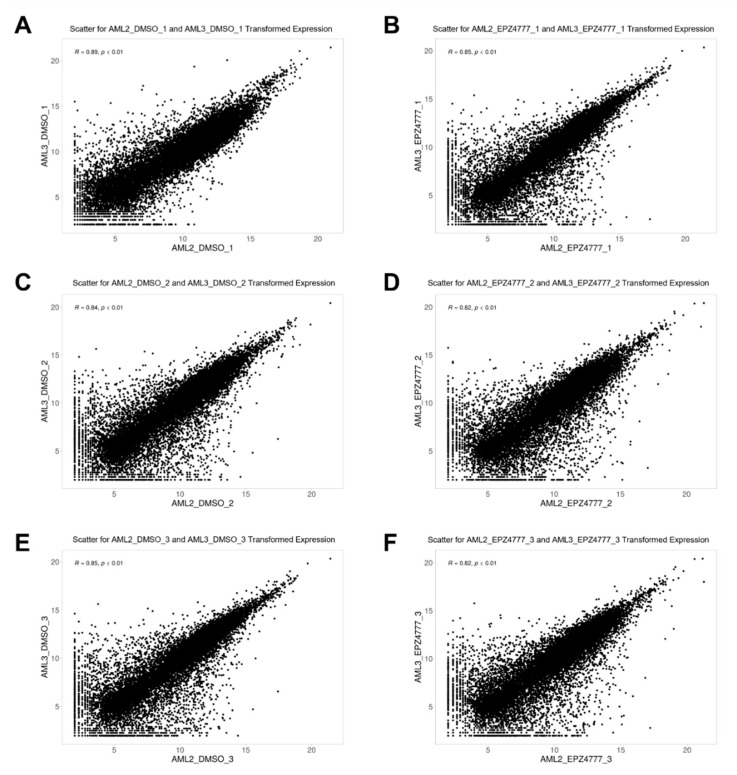
Scatter plots of transformed expression data. (**A**–**F**) The scatter plots show log2-transformed gene expression values for the AML2 and AML3 cell lines treated with DMSO (control) and EPZ004777. Each plot illustrates the correlation of gene expression between different replicates. The R value represents the Pearson correlation coefficient.

**Figure 6 cimb-47-00173-f006:**
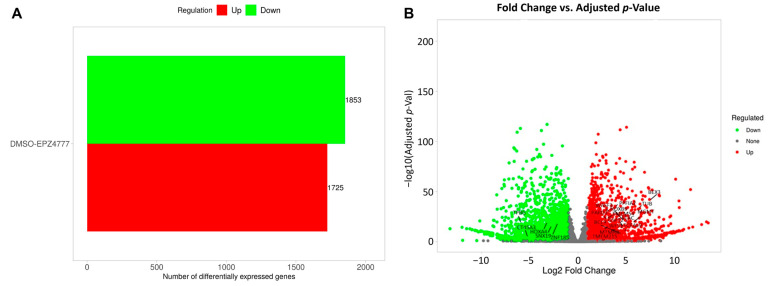
Numbers of DEG2s and Log2 fold change graphs. The experimental design is the cell line and gives the change in the AML3 line compared to the AML2 reference category. (**A**) The comparison between EPZ004777 and DMSO reveals the number of differentially regulated genes, with red indicating upregulated genes (1725 genes) and green indicating downregulated genes (1853 genes). (**B**) A volcano plot illustrates the correlation between log2 fold change and adjusted *p*-value, with green points indicating downregulated genes, red points representing upregulated genes, and gray points denoting genes with no significant change.

**Figure 7 cimb-47-00173-f007:**
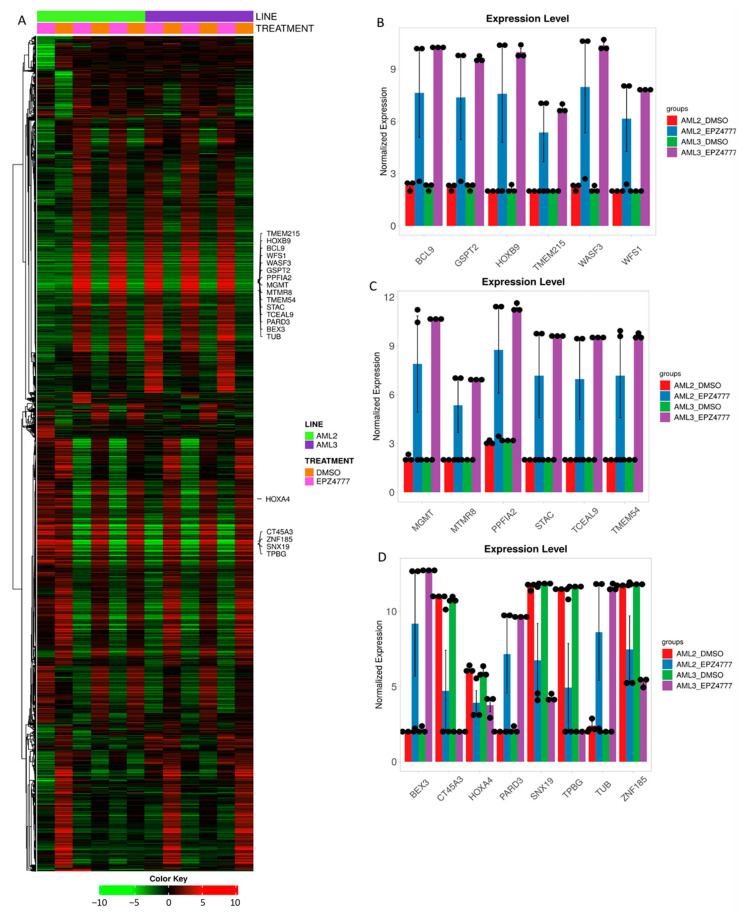
The heatmap and the expression profiles of the significant genes. (**A**) The heatmap shows gene expression levels in the AML2 and AML3 cell lines treated with DMSO or EPZ004777. The color stripes at the top indicate cell lines and treatments, with green for downregulation and red for upregulation. The genes are sorted by expression across conditions. (**B**–**D**) The bar graphs display normalized expression levels of selected genes, with error bars showing standard deviation. The black dots represent biological replicates, and statistical significance is noted where applicable.

**Figure 8 cimb-47-00173-f008:**
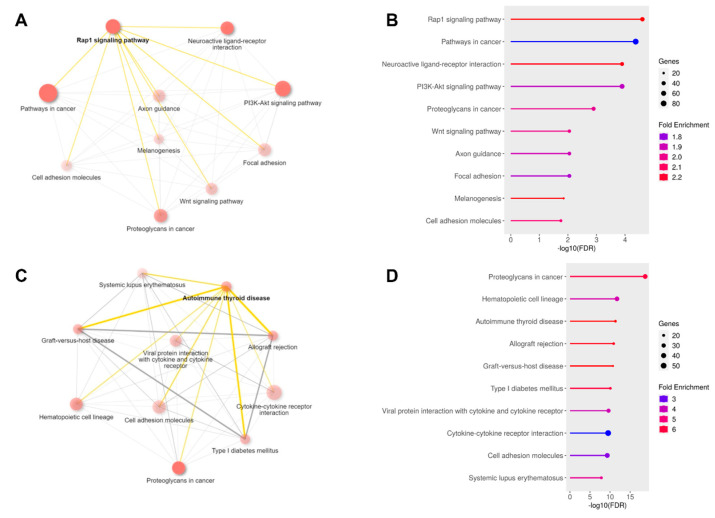
Pathway enrichment analysis of DEGs in AML2 and AML3 treated with EPZ004777. The experimental design is the cell line and gives the change in the AML3 line compared to the AML2 reference category. (**A**) Network plot showing the downregulated pathways enriched in AML2-AML3. The size of the nodes represents pathway significance, and the connections indicate shared genes between pathways. (**B**) Bar plot ranking the top 10 downregulated pathways by −log10(FDR). (**C**) Network plot displaying the upregulated pathways in AML2-AML3. (**D**) Bar plot ranking the top 10 upregulated pathways by −log10(FDR).

**Figure 9 cimb-47-00173-f009:**
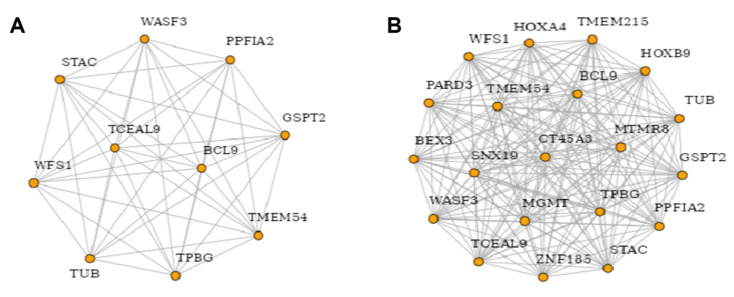
PPI networks of DEGs in AML cell lines treated with EPZ004777 (top 10 and top 20 genes). (**A**) PPI network of selected DEGs including BCL9, WASF3, PPFIA2, GSPT2, WFS1, and TUB. (**B**) An expanded PPI network that includes additional proteins like HOXA4, TMEM215, HOXB9, and MTMR8, illustrating a more intricate interaction network.

**Figure 10 cimb-47-00173-f010:**
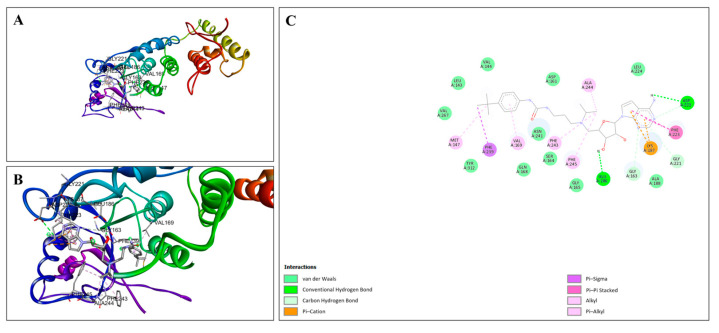
Molecular docking analysis of EPZ004447 with DOT1L. (**A**) Docking pose of EPZ004447 in the DOT1L binding site, showing stable interactions with surrounding amino acid residues. The secondary structure of protein is visualized, and the ligand is shown in stick form. (**B**) Close-up view highlighting specific interactions between EPZ004447 and DOT1L. (**C**) Two-dimensional interaction diagram detailing binding interactions, including hydrogen bonds, van der Waals, Pi–Sigma, Pi–Pi Stacked, alkyl, and Pi–Cation interactions.

**Figure 11 cimb-47-00173-f011:**
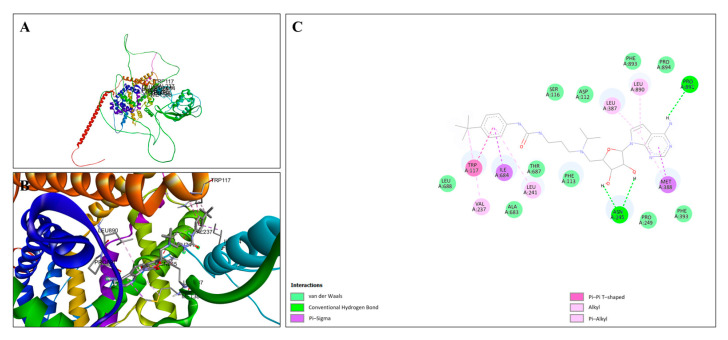
Molecular docking analysis of EPZ004447 with SNX19. (**A**) Docking pose of EPZ004447 in the SNX19 binding site, showing a stable interaction network. The SNX19 protein is visualized with its secondary structure, and the ligand is depicted in stick form. (**B**) Close-up view of the binding site, highlighting specific interactions between EPZ004447 and SNX19. (**C**) Two-dimensional interaction diagram detailing interactions, including hydrogen bonds, van der Waals, Pi–Sigma, Pi–Alkyl, and alkyl interactions.

**Figure 12 cimb-47-00173-f012:**
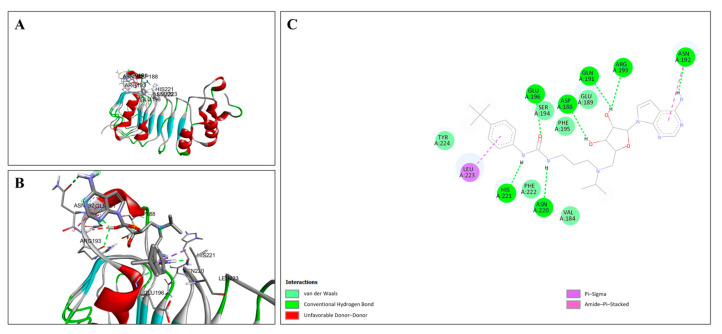
Molecular docking analysis of EPZ004447 with TPBG. (**A**) Docking pose of EPZ004447 in the TPBG binding site, showing stable interactions with surrounding amino acids. The TPBG protein is visualized with its secondary structural elements, and the ligand is shown in stick form. (**B**) Close-up view of the binding site, highlighting specific interactions between EPZ004447 and TPBG. (**C**) Two-dimensional interaction diagram detailing binding interactions, including hydrogen bonds, van der Waals, Pi–Sigma, and Amide–Pi Stacked interactions.

**Figure 13 cimb-47-00173-f013:**
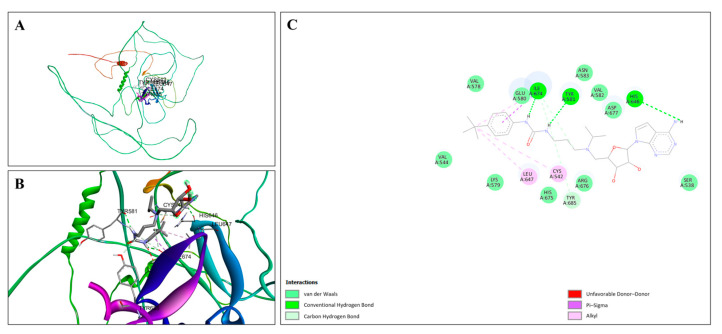
Molecular docking analysis of EPZ004447 with ZNF185. (**A**) Docking pose of EPZ004447 in the ZNF185 binding site, showing stable interactions with surrounding amino acid residues. The ZNF185 protein is visualized with its secondary structural elements, and the ligand is depicted in stick form. (**B**) Close-up view of the binding site, highlighting specific interactions between EPZ004447 and ZNF185. (**C**) Two-dimensional interaction diagram detailing binding interactions, including hydrogen bonds, van der Waals, carbon–hydrogen bonds, Pi–Sigma, and alkyl interactions.

**Table 1 cimb-47-00173-t001:** Experimental design.

Experimental Design	Cell Line	Treatment
AML2_EPZ004777_1	AML2	EPZ004777
AML2_DMSO_1	AML2	DMSO
AML2_EPZ004777_2	AML2	EPZ004777
AML2_DMSO_2	AML2	DMSO
AML2_EPZ004777_3	AML2	EPZ004777
AML2_DMSO_3	AML2	DMSO
AML3_EPZ004777_1	AML3	EPZ004777
AML3_DMSO_1	AML3	DMSO
AML3_EPZ004777_2	AML3	EPZ004777
AML3_DMSO_2	AML3	DMSO
AML3_EPZ004777_3	AML3	EPZ004777
AML3_DMSO_3	AML3	DMSO

**Table 2 cimb-47-00173-t002:** Grid box parameters for molecular docking simulations.

Protein	Center at (X, Y, Z)	Dimension (Å)
BEX3	X: 23.7, Y: 2.969, Z: −16.044	100 Å × 100 Å × 100 Å
CT45A3	X: 9.031, Y: 9.16, Z: 5.757	100 Å × 100 Å × 100 Å
HOXA4	X: −15.665, Y: 5.339, Z: −0.047	60 Å × 60 Å × 60 Å
SNX19	X: 1.419, Y: −1.15, Z: −3.045	60 Å × 60 Å × 60 Å
TPBG	X: 22.829, Y: −1.218, Z: −12.068	40 Å × 40 Å × 40 Å
ZNF185	X: −7.809, Y: 7.241, Z: −1.59	60 Å × 60 Å × 60 Å
DOT1L	X: 29.253, Y: 57.69, Z: 2.739	40 Å × 40 Å × 40 Å

**Table 3 cimb-47-00173-t003:** Gene expression patterns following EPZ004777 treatment based on DEG2 results.

Group	Ensembl Id	Symbol	Entrezgene Id	Log_2_FC Values	Description
Upregulated	ENSG00000132970	WASF3	10810	4.48	WASP family member 3
Upregulated	ENSG00000139220	PPFIA2	8499	4.6	PTPRF interacting protein alpha 2
Upregulated	ENSG00000189369	GSPT2	23708	4.19	G1 to S phase transition 2
Upregulated	ENSG00000121900	TMEM54	113452	4.27	Transmembrane protein 54
Upregulated	ENSG00000166402	TUB	7275	5.25	TUB bipartite transcription factor
Upregulated	ENSG00000109501	WFS1	7466	3.24	Wolframin ER transmembrane glycoprotein
Upregulated	ENSG00000144681	STAC	6769	4.32	SH3 and cysteine rich domain
Upregulated	ENSG00000185222	TCEAL9	51186	4.4	Transcription elongation factor A like 9
Upregulated	ENSG00000116128	BCL9	607	4.7	BCL9 transcription coactivator
Upregulated	ENSG00000188133	TMEM215	401498	2.5	Transmembrane protein 215
Upregulated	ENSG00000170689	HOXB9	3219	4.31	Homeobox B9
Upregulated	ENSG00000166681	BEX3	27018	6.25	Brain expressed X-linked 3
Upregulated	ENSG00000148498	PARD3	56288	4.44	Par-3 family cell polarity regulator
Upregulated	ENSG00000170430	MGMT	4255	5.01	O-6-methylguanine-DNA methyltransferase
Upregulated	ENSG00000102043	MTMR8	55613	2.99	Myotubularin related protein 8
Downregulated	ENSG00000197576	HOXA4	3201	−4.3	Homeobox A4
Downregulated	ENSG00000147394	ZNF185	7739	−3.66	Zinc finger protein 185
Downregulated	ENSG00000146242	TPBG	7162	−5.53	Trophoblast Glycoprotein
Downregulated	ENSG00000120451	SNX19	399979	−4.1	Sorting nexin 19
Downregulated	ENSG00000269096	CT45A3	441519	−5.12	Cancer/testis antigen family 45 member A3

**Table 4 cimb-47-00173-t004:** Cell signaling pathways clustered based on upregulated and downregulated genes following EPZ004777 treatment according to the DEG2 results.

Direction	DEG2 Analysis: AML2 vs. AML3 Pathways	Fold Enriched	nGenes	−log10 (FDR)
Down	Rap1 signaling pathway	2.279	43	4.8
	Cell adhesion molecules	2.112	27	1.97
	Neuroactive ligand–receptor interaction	2.245	35	3.61
	Melanogenesis	2.336	19	2.0
	Proteoglycans in cancer	2.038	38	3.18
Up	Autoimmune thyroid disease	5.631	14	12.8
	Graft-versus-host disease	5.456	13	12.6
	Allograft rejection	5.406	14	11.9
	Type I diabetes mellitus	5.005	14	10.8
	Proteoglycans in cancer	5.019	28	17.5

FDR: False Discovery Rate, −log10(FDR): Negative logarithm (base 10) of the False Discovery Rate (used to show statistical significance).

**Table 5 cimb-47-00173-t005:** Molecular docking results of EPZ004777 with selected target proteins.

Protein	Minimum Binding Energy (Kcal/mol)	Key Structural Amino Acid Residues	Hydrogen Bond Distance (Å)	Number of Hydrogen Bonds
DOT1L	−9.7	ASP222	2.74	5
			3.58	
		GLY163	3.62	
		GLU186	2.28	
		GLY221	3.63	
SNX19	−8.2	PRO891	2.77	3
		ASN245	2.41	
			2.81	
TPBG	−7.7	GLU196	2.11	7
		ASN192	2.60	
		GLN191	2.02	
		ARG193	2.45	
		ASP188	2.35	
		ASN220	2.34	
		HIS221	1.86	
ZNF185	−7.2	HIS646	2.35	6
		TYR581	2.10	
		ILE674	2.18	
			3.53	
			3.50	
		TYR685	3.48	
CT45A3	−6.7	HIS180	2.48	2
			3.36	
HOXA4	−6.5	LYS199	2.49	2
		LYS200	2.98	
BEX3	−5.2	ASN46	2.01	4
		PHE47	2.63	
			2.77	
		ALA44	3.24	

## Data Availability

The data are available from the authors on reasonable request.

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
