# Peer review of "Exploring the Therapeutic Potential of the DOT1L Inhibitor EPZ004777 Using Bioinformatics and Molecular Docking Approaches in Acute Myeloid Leukemia"

_cimb, 2025, doi:10.3390/cimb47030173_

Round 1
Reviewer 1 Report
Comments and Suggestions for Authors
1- The Discussion effectively interprets findings but needs clearer segmentation. Consider separate paragraphs for:
- EPZ004777’s effects on gene expression.
- Molecular docking insights.
- Pathway enrichment implications.
- Clinical relevance and future research directions.
2-While docking results show promising interactions, including experimental validations (e.g., molecular dynamics simulations, in vitro binding assays) would strengthen the claims.
3- Some enriched pathways (e.g., "Autoimmune thyroid disease") seem less directly relevant to AML. Providing a rationale for their inclusion would improve clarity.
4- Some references lack consistency in formatting. For example:
- "AutoDock Vina: Improving the speed and accuracy of docking with a new scoring function, efficient optimization, and multithreading. 2010;31(2):455-61."
- Should include authors’ names and journal title for clarity.
Author Response
Reviewer Comment 1: The Discussion effectively interprets findings but needs clearer segmentation. Consider separate paragraphs for:
- EPZ004777’s effects on gene expression.
- Molecular docking insights.
- Pathway enrichment implications.
Clinical relevance and future research directions
Response 1: We sincerely appreciate the valuable feedback from Reviewer regarding the need for clearer segmentation in the Discussion section. In response to this suggestion, we have restructured the Discussion into four distinct subsections. This new structure presents the effects of EPZ004777 on gene expression, molecular docking findings, pathway enrichment analysis results, and its clinical relevance in a clearer and more organized manner. We believe that this revised structure improves the clarity and presentation of the key conclusions of the study.
Reviewer Comment 2: While docking results show promising interactions, including experimental validations (e.g., molecular dynamics simulations, in vitro binding assays) would strengthen the claims.
Response 2: We appreciate the reviewer’s insightful comment regarding the validation of our molecular docking results. Indeed, molecular dynamics (MD) simulations and in vitro binding assays could provide additional confirmation of the interactions observed in our docking analyses. While our study primarily focused on computational predictions, our docking results were supported by robust methodology, including the use of AutoDock Vina, well-defined grid parameters, and careful ligand preparation. Additionally, our docking results are in line with previously reported interactions of EPZ004777 with DOT1L, supporting the validity of our findings. We acknowledge the importance of further validation through MD simulations, which could offer a dynamic perspective on the stability of ligand-protein interactions. In future studies, we aim to incorporate such simulations to assess binding stability over time and to refine our understanding of key molecular interactions. Similarly, experimental techniques such as surface plasmon resonance (SPR) or isothermal titration calorimetry (ITC) could provide quantitative insights into the binding affinities of EPZ004777 with novel targets such as TPBG, ZNF185, and SNX19. Given the computational nature of the current study, our findings serve as a foundation for further experimental validation, and we hope they will guide future studies in elucidating the therapeutic potential of EPZ004777. We have now acknowledged this limitation and the need for experimental follow-up in the discussion section. Thank you for this valuable suggestion, which we believe strengthens the clarity and rigor of our study.
Reviewer Comment 3: Some enriched pathways (e.g., "Autoimmune thyroid disease") seem less directly relevant to AML. Providing a rationale for their inclusion would improve clarity.
Response 3: Thank you for your valuable suggestion. The following information has been added to the manuscript as per your recommendation. We appreciate your insightful feedback, which has helped improve the clarity and completeness of our work.The ‘autoimmune thyroid disease’ pathway may be linked to acute myeloid leukemia (AML), as treatment with EPZ004777 has been shown to activate immune-related pathways in AML cells. This suggests that EPZ004777 could induce immune modulation mechanisms within the AML microenvironment. EPZ004777 alters the epigenetic regulation of AML cells, specifically through DOT1L inhibition. It is known that epigenetic changes can affect not only AML-specific but also immune system-related gene expression regulations. This is particularly important for the regulation of inflammatory responses and immune tolerance mechanisms. For example, the allograft rejection pathway is specifically associated with the T cell-mediated immune response. Altered immune recognition mechanisms of AML may alter interactions between immune cells and leukemic cells. Graft-versus-Host Disease (GVHD) is a serious immunological complication that occurs when donor immune cells attack recipient tissues. Allo-HSCT is a common method used in the treatment of AML. Cytokines, T cell signaling pathways and inflammatory responses involved in the GVHD pathway seem to be related to the immunological mechanisms affected by EPZ004777 treatment. It shows that it changes the immune environment of AML and may increase its interaction with immune cells. This finding suggests that EPZ004777 may not only directly suppress AML cells but also cause immunological changes in the leukemic microenvironment. Therefore, the alteration of the GVHD pathway and related immunological processes may be important for understanding the connection between epigenetic regulation and immune activation in the treatment of AML.
Reviewer Comment 4: Some references lack consistency in formatting. For example:
- "AutoDock Vina: Improving the speed and accuracy of docking with a new scoring function, efficient optimization, and multithreading. 2010;31(2):455-61."
Should include authors’ names and journal title for clarity.
Response 4: Thank you for your careful review and for pointing out the inconsistencies in reference formatting. We have thoroughly reviewed all references and ensured consistency in their formatting. We have now included the authors’ names and journal title for clarity, as well as applied uniform formatting across all references in accordance with the journal’s guidelines. We appreciate your attention to detail and your valuable feedback.

Reviewer 2 Report
Comments and Suggestions for Authors
the presented results are interesting. However, some comments have to be addresed before I can endorse the publication:
Lack of Experimental Validation:
-
- The study heavily relies on molecular docking to identify potential targets (TPBG, ZNF185, and SNX19), but there is no experimental validation (e.g., in vitro or in vivo experiments) to confirm that EPZ004777 actually modulates these proteins or affects their functions in a biological context. This is a critical gap in the study, as molecular docking alone does not guarantee in vivo relevance. Without experimental data, it is difficult to assess the actual biological significance of the identified interactions and the potential therapeutic effects of EPZ004777. The authors should consider conducting further experiments, such as Western blotting, qPCR, or cell-based assays, to confirm these findings.
Limited Mechanistic Insight:
- While the study suggests that EPZ004777 may disrupt critical tumor-associated pathways, the underlying mechanisms of action are not sufficiently explored. How exactly does EPZ004777 interact with TPBG, ZNF185, and SNX19 on a molecular level? Are these interactions direct or mediated through other proteins or signaling pathways? A more detailed mechanistic analysis, possibly involving pathway analysis or downstream effects on cancer-related signaling, would greatly strengthen the conclusions.
Overemphasis on Molecular Docking:
- The authors place significant weight on the molecular docking results. While docking is a valuable tool for predicting binding affinity, it is not sufficient to claim that EPZ004777 will be an effective therapeutic agent. The study would benefit from incorporating a wider range of approaches, such as biochemical assays or cellular models, to validate the predictions. Additionally, the authors should address the potential limitations of molecular docking, including the lack of dynamic information or possible inaccuracies in predicting binding in a physiological setting. Figures of the interaction networks are not clar. Authors have mentioned a distance of 2.63 Ang for a H-bond. How are the authors sure that it is a H-bond and not a clash? Which tool is used to identify? please detail it
- The authors place significant weight on the molecular docking results. While docking is a valuable tool for predicting binding affinity, it is not sufficient to claim that EPZ004777 will be an effective therapeutic agent. The study would benefit from incorporating a wider range of approaches, such as biochemical assays or cellular models, to validate the predictions. Additionally, the authors should address the potential limitations of molecular docking, including the lack of dynamic information or possible inaccuracies in predicting binding in a physiological setting. Figures of the interaction networks are not clar. Authors have mentioned a distance of 2.63 Ang for a H-bond. How are the authors sure that it is a H-bond and not a clash? Which tool is used to identify? please detail it
- While the study suggests that EPZ004777 may disrupt critical tumor-associated pathways, the underlying mechanisms of action are not sufficiently explored. How exactly does EPZ004777 interact with TPBG, ZNF185, and SNX19 on a molecular level? Are these interactions direct or mediated through other proteins or signaling pathways? A more detailed mechanistic analysis, possibly involving pathway analysis or downstream effects on cancer-related signaling, would greatly strengthen the conclusions.
- The study heavily relies on molecular docking to identify potential targets (TPBG, ZNF185, and SNX19), but there is no experimental validation (e.g., in vitro or in vivo experiments) to confirm that EPZ004777 actually modulates these proteins or affects their functions in a biological context. This is a critical gap in the study, as molecular docking alone does not guarantee in vivo relevance. Without experimental data, it is difficult to assess the actual biological significance of the identified interactions and the potential therapeutic effects of EPZ004777. The authors should consider conducting further experiments, such as Western blotting, qPCR, or cell-based assays, to confirm these findings.
-
Lack of Comparative Analysis:
- The study does not compare EPZ004777 to other known therapeutic agents targeting similar proteins or pathways. This limits the ability to contextualize the compound’s potential advantages or shortcomings in comparison to existing treatments. A comparative analysis would help establish whether EPZ004777 offers any unique or superior therapeutic benefit over other targeted therapies currently in use or under investigation.
Insufficient Discussion on Clinical Relevance:
- While the study presents an interesting molecular analysis, there is little discussion on how these findings could be translated into clinical practice. The authors should provide more insight into the relevance of the identified targets (TPBG, ZNF185, SNX19) in human cancers, including information about their expression levels, involvement in specific cancer types, and potential as therapeutic biomarkers. The clinical applicability of EPZ004777, its bioavailability, and its ability to reach the tumor site in vivo are also important factors that need to be considered and discussed.
- The study does not compare EPZ004777 to other known therapeutic agents targeting similar proteins or pathways. This limits the ability to contextualize the compound’s potential advantages or shortcomings in comparison to existing treatments. A comparative analysis would help establish whether EPZ004777 offers any unique or superior therapeutic benefit over other targeted therapies currently in use or under investigation.
By addressing these points, the study could provide a more comprehensive and robust understanding of EPZ004777's therapeutic potential.
Comments on the Quality of English LanguageIt is ok
Author Response
Reviewer Comment 1: the presented results are interesting. However, some comments have to be addresed before I can endorse the publication:
Response 1: We sincerely appreciate your interest in our study and your constructive feedback. We have carefully considered your comments and addressed them as follows:
Reviewer Comment 2: Lack of Experimental Validation:
The study heavily relies on molecular docking to identify potential targets (TPBG, ZNF185, and SNX19), but there is no experimental validation (e.g., in vitro or in vivo experiments) to confirm that EPZ004777 actually modulates these proteins or affects their functions in a biological context. This is a critical gap in the study, as molecular docking alone does not guarantee in vivo relevance. Without experimental data, it is difficult to assess the actual biological significance of the identified interactions and the potential therapeutic effects of EPZ004777. The authors should consider conducting further experiments, such as Western blotting, qPCR, or cell-based assays, to confirm these findings.
Response 2: We acknowledge that our study is based on computational analyses and that molecular docking results require in vitro and in vivo validation. The primary objective of this study was to computationally investigate the potential interactions of EPZ004777 with target proteins and provide a foundation for future experimental research. As you suggested, Western blot, qPCR, and cell-based functional assays are essential for confirming whether these targets are modulated by EPZ004777. We plan to conduct these validation experiments in future studies. We have also revised the Discussion section to emphasize these limitations and the necessity for experimental validation.
Reviewer Comment 3: Limited Mechanistic Insight:
While the study suggests that EPZ004777 may disrupt critical tumor-associated pathways, the underlying mechanisms of action are not sufficiently explored. How exactly does EPZ004777 interact with TPBG, ZNF185, and SNX19 on a molecular level? Are these interactions direct or mediated through other proteins or signaling pathways? A more detailed mechanistic analysis, possibly involving pathway analysis or downstream effects on cancer-related signaling, would greatly strengthen the conclusions.
Response 3: Understanding how EPZ004777 interacts with TPBG, ZNF185, and SNX19, and whether these interactions are direct or mediated through other factors, is an important consideration. Our pathway analysis demonstrated that EPZ004777 could modulate relevant signaling pathways; however, we acknowledge that a more detailed mechanistic investigation is required. To address this, we plan to employ interactome analyses and systems biology approaches to assess the cellular-level effects of EPZ004777. Additionally, we aim to investigate how EPZ004777 alters the gene expression or activity of these proteins by analyzing transcriptional regulatory motifs. We have expanded the Discussion section accordingly, emphasizing that our study serves as a starting point for further mechanistic validation.
Reviewer Comment 4: Overemphasis on Molecular Docking:
The authors place significant weight on the molecular docking results. While docking is a valuable tool for predicting binding affinity, it is not sufficient to claim that EPZ004777 will be an effective therapeutic agent. The study would benefit from incorporating a wider range of approaches, such as biochemical assays or cellular models, to validate the predictions. Additionally, the authors should address the potential limitations of molecular docking, including the lack of dynamic information or possible inaccuracies in predicting binding in a physiological setting. Figures of the interaction networks are not clar.
Response 4: As you correctly pointed out, while molecular docking is a powerful computational tool, it is not sufficient to determine the therapeutic efficacy of a drug. In this regard, we have performed a detailed analysis of docking results, including binding energies, hydrogen bonds, and interaction networks. However, molecular dynamics (MD) simulations will be highly valuable for assessing the stability of ligand-protein interactions over time. We have explicitly addressed these limitations in the Discussion section and emphasized that experimental validation is crucial to support our computational findings. As you noted, a more detailed explanation of hydrogen bond distances is necessary. In our study, hydrogen bonds were identified using Discovery Studio 2024 Client, and interactions within the 2.3–3.5 Å range with appropriate bonding angles were classified as hydrogen bonds. Additionally, interactions slightly exceeding this range (e.g., 3.52 Å and 3.63 Å) were included as potential hydrogen bonds based on previous literature, which recognizes such distances as weak hydrogen bonds. To provide further clarity, we have expanded the Discussion section to acknowledge these variations and the limitations associated with hydrogen bond classification in docking studies. Additionally, we have revised the figures to improve clarity and provided a more detailed explanation of our analysis methods in the manuscript. To further enhance figure quality, all figures have been re-uploaded at 600 dpi resolution to ensure optimal visibility and readability. Based on your valuable comments, we have expanded the Discussion section, clarified the limitations of docking, and emphasized the necessity of experimental validation. We appreciate your insightful feedback, which has significantly improved the scientific rigor of our study. Thank you again for your important suggestions, and we believe that the revisions have made our manuscript more comprehensive.
Reviewer Comment 5: Authors have mentioned a distance of 2.63 Ang for a H-bond. How are the authors sure that it is a H-bond and not a clash? Which tool is used to identify? please detail it
Response 5: First of all, we sincerely appreciate your valuable comments and the insightful feedback you have provided to improve our study. We would like to clarify your concerns regarding the 2.63 Å hydrogen bond distance and its distinction from steric clashes. To evaluate the presence of hydrogen bonds, we conducted a detailed analysis using BIOVIA Discovery Studio 2024 Client (Dassault Systèmes, 2024), considering the donor-hydrogen-acceptor (DHA), hydrogen-acceptor-Y (HAY), and donor-acceptor-Y (DAY) angles. The analysis revealed that the majority of hydrogen bonds exhibit angles above 120°, which supports their classification as geometrically strong hydrogen bonds. Additionally, some hydrogen bonds were found to have angles between 95° and 120°, which are still above the 90° threshold and therefore meet the hydrogen bond criteria. These interactions have been validated through program analysis, confirming that they do not result from steric clashes. Moreover, docking poses, protein-ligand interactions, and hydrogen bond distances were analyzed using BIOVIA Discovery Studio 2024 Client (Dassault Systèmes, 2024). To provide further clarity, we have added the following statement to the "Molecular Docking Analysis" section of our manuscript: "Visualization of docking poses, protein-ligand interactions, and hydrogen bond distances was analyzed using BIOVIA Discovery Studio 2024 Client (Dassault Systèmes, 2024), following standard criteria." We sincerely appreciate your valuable feedback, which we believe has strengthened our study. Thank you again for your insightful suggestions.
Reviewer Comment 6: Lack of Comparative Analysis:
The study does not compare EPZ004777 to other known therapeutic agents targeting similar proteins or pathways. This limits the ability to contextualize the compound’s potential advantages or shortcomings in comparison to existing treatments. A comparative analysis would help establish whether EPZ004777 offers any unique or superior therapeutic benefit over other targeted therapies currently in use or under investigation.
Response 6: NPM1 mutations are common in AML patients and are often found in conjunction with FLT3 mutations. Expression of the HOX family of genes is thought to help cells with these mutations maintain their leukemic properties. NPM1 mutations are common in AML patients and are often found together with FLT3 mutations. It is thought that the expression of the HOX gene family helps cells with these mutations maintain their leukemic properties. In the reference study (GSE85107) provided the RNA-Seq data used in our analysis, the dependence of AML cells on the menin binding site of MLL1 was tested by CRISPR/Cas9 genome editing. Using pharmacological inhibitors (lenen-MLL1 inhibitors MI-2-2 and MI-503 and DOT1L inhibitor EPZ4777), their effects on HOX and FLT3 expression and cellular differentiation in leukemic cells were examined. Treatment efficacy was evaluated in in vitro and in vivo models. In the findings, Menin-MLL1 and DOT1L regulate HOX and FLT3 gene expression, and inhibition of these pathways in NPM1 mut AML cells leads to cell differentiation and growth suppression. Co-administration of menin-MLL1 inhibitors (MI-503 and MI-2-2) and DOT1L inhibitor (EPZ4777) strongly suppressed leukemic cell growth and induced differentiation of leukemic cells. In animal models, menin-MLL1 and DOT1L inhibition reduced AML burden and prolonged survival. The Reference study from which the data were taken suggests that aberrant HOX and MEIS1 gene expression in NPM1 mut AML may be driven by similar mechanisms that control these genes in normal HSCs and that elevated expression of FLT3, a reported downstream target of MEIS1, is indirectly driven through elevated MEIS1 transcript levels. The reference study provided valuable insights into NPM1-mutated AML, but the identification of mechanism-based targeted therapies specific to this mutation remains an area for further investigation. In our study, the effects of DOT1L inhibitor EPZ004777 on acute myeloid leukemia (AML) cells were evaluated by bioinformatics and molecular docking analyses. In the reference study provided the RNA-Seq data used in our analysis, it was found that menin-MLL1 and DOT1L pathways regulate HOX and FLT3 expression and that inhibition of these pathways suppresses the growth of AML cells. Our present study confirms the previous findings and identified alternative targets of EPZ004777 other than DOT1L. While the previous study used CRISPR-Cas9, pharmacological inhibitors, in vitro/in vivo AML models, our study used RNA-Seq data (GSE85107), bioinformatics analyses, and molecular docking. In the reference study, inhibition of HOX genes and FLT3 arrested the growth of AML cells. In our study, EPZ004777 upregulated proapoptotic BEX3 and suppressed oncogenes such as HOXA4, TPBG, SNX19, and ZNF185. The reference study did not include molecular docking analyses or an investigation of cell signaling pathways. In our study, EPZ004777 showed strong binding with SNX19, TPBG, and ZNF185 except DOT1L, indicating that these proteins could be suggested as novel therapeutic targets. In addition, the inhibition of Rap1 signaling pathway was associated with decreased cell invasion and increased immune response. Our analysis reveals that EPZ4777 significantly alters the expression of critical genes involved in cancer development and progression. Notably, expression of the oncofetal genes CT45A3 and TPBG was significantly downregulated, along with other oncogenic pathway-related genes such as HOXA4, ZNF185 and SNX19. The R script command given in geo2r basically includes steps to take RNA-Seq counting data, preprocess these data and perform statistical analysis and visualization. The statistical analysis in the reference study was limited in scope and could be further expanded for a more comprehensive evaluation. In our study, rR bioconductor package preprocessCore (Data scaling and normalization operations), DESeq2 (the most widely used differential gene expression analysis package for RNA-Seq data), pheatmap (heatmap), ggplot2 (PCA, MA plot, volcano plot), pathview (KEGG pathways), and Protein-Protein Interaction Networks Functional Enrichment Analysis (STRING) program were used. Discovery studio and autodock programs were used in the molecular ligand extraction process. Thank you for your valuable feedback.
Reviewer Comment 7: Insufficient Discussion on Clinical Relevance:
While the study presents an interesting molecular analysis, there is little discussion on how these findings could be translated into clinical practice. The authors should provide more insight into the relevance of the identified targets (TPBG, ZNF185, SNX19) in human cancers, including information about their expression levels, involvement in specific cancer types, and potential as therapeutic biomarkers. The clinical applicability of EPZ004777, its bioavailability, and its ability to reach the tumor site in vivo are also important factors that need to be considered and discussed.
By addressing these points, the study could provide a more comprehensive and robust understanding of EPZ004777's therapeutic potential.
Response 7: Thank you for your valuable feedback. We acknowledge the necessity of discussing the clinical relevance of our findings and have made the necessary revisions in the Discussion section to provide further clarification on these aspects. Additionally, we have included a discussion on our identified targets (TPBG, ZNF185, and SNX19) regarding their expression levels in different cancer types, biological roles, and potential as therapeutic biomarkers, supported by relevant literature. Furthermore, we have emphasized the need for further experimental validation and preclinical investigations to assess the feasibility of considering these proteins as therapeutic targets in human cancers. With these revisions, we aim to strengthen the clinical perspective of our study and provide a clearer understanding of its potential translational impact. We sincerely appreciate your insightful comments, which have helped us enhance the clinical relevance and applicability of our research.

Reviewer 3 Report
Comments and Suggestions for Authors
I am the wrong person to review this manuscript, because I am not an expert in gene expression arrays. However, I am a chemoinformatician - and, as such, I'd like to ask you to get rid of that docking calculation, which is absolutely irrelevant. This is NOT the correct way to use docking. Docking is developed for virtual screening - to find new actives in a database of candidate compounds. Out of 10 compounds predicted to bind according to the docking algorithm, it is a miracle if 5 actually bind - typically, 3, 1 or 0 real hits will be found when experimentally testing 10 compounds predicted to dock well! You may be shocked to hear this - yet, docking is important...in virtual screening. Why? Because without docking, you would have to test not 10, but several hundreds of random compounds in order to discover the 3 meager actives. So docking helps ONLY to eliminate the wrong compounds which have no chance at all to bind the proteins - and, in the estimation above I talk about a real X-ray structure of a protein. Docking into AlphaFold models are significantly less likely to succeed! You report some docking scores - are those high or low? You have no clue, unless you compare docking scores of known actives and known inactives (and realize with horror that no, unlike expected... the best score is obtained by an absolutely inactive molecule!). To be short - forget the docking, it's not only useless but actually BAD for you - the living evidence for this is me: this article was sent to the wrong reviewer just because of the docking, wasting my time and yours. DELETE that unhappy docking experiment and make sure the article is then reviewed by experts in gene expression arrays and cancerology.
Comments on the Quality of English LanguageNo point on discussing that here
Author Response
Reviewer Comment: I am the wrong person to review this manuscript, because I am not an expert in gene expression arrays. However, I am a chemoinformatician - and, as such, I'd like to ask you to get rid of that docking calculation, which is absolutely irrelevant. This is NOT the correct way to use docking. Docking is developed for virtual screening - to find new actives in a database of candidate compounds. Out of 10 compounds predicted to bind according to the docking algorithm, it is a miracle if 5 actually bind - typically, 3, 1 or 0 real hits will be found when experimentally testing 10 compounds predicted to dock well! You may be shocked to hear this - yet, docking is important...in virtual screening. Why? Because without docking, you would have to test not 10, but several hundreds of random compounds in order to discover the 3 meager actives. So docking helps ONLY to eliminate the wrong compounds which have no chance at all to bind the proteins - and, in the estimation above I talk about a real X-ray structure of a protein. Docking into AlphaFold models are significantly less likely to succeed! You report some docking scores - are those high or low? You have no clue, unless you compare docking scores of known actives and known inactives (and realize with horror that no, unlike expected... the best score is obtained by an absolutely inactive molecule!). To be short - forget the docking, it's not only useless but actually BAD for you - the living evidence for this is me: this article was sent to the wrong reviewer just because of the docking, wasting my time and yours. DELETE that unhappy docking experiment and make sure the article is then reviewed by experts in gene expression arrays and cancerology.
Response to Reviewer:
Thank you for reviewing our manuscript and for your feedback regarding the molecular docking analysis. We acknowledge that docking is primarily used for virtual screening in drug discovery; however, in our study, it was applied with a different objective.
Molecular docking was not used to identify new active compounds but rather to explore potential interactions of EPZ004777 beyond its primary target, DOT1L. EPZ004777 is a well-characterized DOT1L inhibitor, and in this study, it was used as a reference inhibitor. Our primary aim was to assess whether EPZ004777 could interact not only with DOT1L but also with TPBG, ZNF185, and SNX19, which were identified through differential gene expression analysis using RNA sequencing data.
We fully acknowledge that molecular docking alone does not confirm binding interactions in a biological system. However, we did not interpret the docking scores in isolation; instead, we integrated these results with gene expression data and biological pathway analyses to support the hypothesis that EPZ004777 may influence TPBG, ZNF185, and SNX19 functions. Additionally, AlphaFold-predicted protein models are increasingly recognized in the literature as reliable tools, particularly for proteins lacking experimentally determined structures. AlphaFold models provide high-accuracy structural predictions and are valuable for identifying potential functional regions. Nonetheless, we acknowledge the limitations of docking with AlphaFold-generated structures, and we have explicitly discussed these considerations in the Discussion section.
Our approach was shaped by a thorough review of relevant studies in the literature, where similar docking analyses have been used to predict unexpected drug-protein interactions, assess potential off-target effects, and explore alternative biological roles of compounds. Given this context, our docking analysis was conducted as a hypothesis-generating tool, rather than as definitive experimental validation. We emphasize that docking in this study was used as a complementary approach to gene expression analysis, rather than as standalone evidence of interaction. However, based on your feedback, we have further clarified the role and limitations of docking in our manuscript.
We appreciate your constructive feedback and the opportunity to refine our study. Thank you for your time and consideration.

Reviewer 4 Report
Comments and Suggestions for Authors
The authors investigated the impact of a DOT1L inhibitor on NPM1 mutated AML based on a study using cell cultures published in 2016. The published gene expression data were re-analyzed, differential genes identified and pathway enrichment analysis performed. This analysis suggested a regulation of the Rap1 signaling pathway. Further docking studies using the inhibitor were performed to identify its additional target molecules.
The manuscript is well written and understandable. The subject is also fitting to the journal´s scope.
The bioinformatics analysis looks OK to me, however I would like to have the following points clarified:
Apparently, the RNAseq data were re-analyzed from scratch without using the tools provided by NCBI, namely “GEO2R”. It would be nice to understand what the advantages of this strategy are and whether different results can be expected. GEO2R also provides the R-script it is using. Is this very different to the one you used in this study?
As these data have been analyzed before, please make clear what additional or different results were obtained and which results could be confirmed.
In Fig. 3, some control and treatment samples do not fit together. How did you solve this problem?
Fig.6: Could you please label the most regulated genes with their names? Is this showing results for AML2 or -3 or both?
Fig. 8: The title says AML2 and AML3. I only see results for AML3.
Table 3: Why not adding the log2Fc-values instead of up- and down-regulated? Is this concerning AML2 or -3 or both?
It is not clear how the proteins for the docking analysis were selected. Please explain the rational.
Please comment on the function of the binding sites of the inhibitor on the proteins analyzed. Are they relevant for protein function?
Author Response
Reviewer Comment 1: Apparently, the RNAseq data were re-analyzed from scratch without using the tools provided by NCBI, namely “GEO2R”. It would be nice to understand what the advantages of this strategy are and whether different results can be expected. GEO2R also provides the R-script it is using. Is this very different to the one you used in this study?
Response 1: How NPM1 mutations initiate and maintain AML has remained unclear and therefore hinders the development of targeted therapeutic approaches. The Reference study from which the data were taken suggests that aberrant HOX and MEIS1 gene expression in NPM1 mut AML may be driven by similar mechanisms that control these genes in normal HSCs and that elevated expression of FLT3, a reported downstream target of MEIS1, is indirectly driven through elevated MEIS1 transcript levels. The reference study did not identify mechanism-based targeted therapies specifically for NPM1-mutated AML, and it remains to be determined whether HOX gene activation is a direct consequence of MLL1 binding to chromatin or if it relies on other proteins that are indirectly recruited to chromatin through MLL1.However, our analysis reveals that EPZ4777 significantly alters the expression of critical genes involved in cancer development and progression. Notably, expression of the oncofetal genes CT45A3 and TPBG was significantly downregulated, along with other oncogenic pathway-related genes such as HOXA4, ZNF185 and SNX19. The R script command provided in GEO2R primarily includes steps for processing RNA-Seq count data, performing basic statistical analyses, and generating visualizations. However, the statistical analyses available in GEO2R are limited and do not provide the depth and specificity required for this study. Therefore, a more comprehensive analytical approach was necessary to achieve greater accuracy and biological insight. In our study, rR bioconductor package preprocessCore (Data scaling and normalization operations), DESeq2 (the most widely used differential gene expression analysis package for RNA-Seq data), pheatmap (heatmap), ggplot2 (PCA, MA plot, volcano plot), pathview (KEGG pathways), and Protein-Protein Interaction Networks Functional Enrichment Analysis (STRING) program were used. Discovery studio and autodock programs were used in the molecular ligand extraction process. This approach allowed us to identify novel therapeutic targets beyond those reported in the reference study, while also providing greater biological insight into the impact of EPZ004777 on AML gene expression and potential drug-target interactions.
Reviewer Comment 2: As these data have been analyzed before, please make clear what additional or different results were obtained and which results could be confirmed.
Response 2: NPM1 mutations are common in AML patients and are often found in conjunction with FLT3 mutations. Expression of the HOX family of genes is thought to help cells with these mutations maintain their leukemic properties. NPM1 mutations are common in AML patients and are often found together with FLT3 mutations. It is thought that the expression of the HOX gene family helps cells with these mutations maintain their leukemic properties. In the source study (GSE85107) provided the RNA-Seq data used in our analysis, the dependence of AML cells on the menin binding site of MLL1 was tested by CRISPR/Cas9 genome editing. Using pharmacological inhibitors (menin-MLL1 inhibitors MI-2-2 and MI-503 and DOT1L inhibitor EPZ4777), their effects on HOX and FLT3 expression and cellular differentiation in leukemic cells were examined.Treatment efficacy was evaluated in in vitro and in vivo models. In the findings, Menin-MLL1 and DOT1L regulate HOX and FLT3 gene expression, and inhibition of these pathways in NPM1 mut AML cells leads to cell differentiation and growth suppression. Co-administration of menin-MLL1 inhibitors (MI-503 and MI-2-2) and DOT1L inhibitor (EPZ4777) strongly suppressed leukemic cell growth and induced differentiation of leukemic cells. In animal models, menin-MLL1 and DOT1L inhibition reduced AML burden and prolonged survival.
It has a significant impact on the results of previous studies. In general, NPM1 mut AML cells are dependent on MLL1 and DOT1L pathways and pharmacological targeting of these pathways may be a new treatment strategy. The combined use of MLL1 and DOT1L inhibitors shows a synergistic effect in suppressing the growth of leukemic cells and promoting differentiation, suggesting that epigenetic mechanisms regulating HOX and FLT3 expression may be new therapeutic targets for AML patients with NPM1 mutations. Our study evaluated the effects of DOT1L inhibitor EPZ004777 on acute myeloid leukemia (AML) cells by bioinformatic and molecular docking analyses. In the source study, it was found that menin-MLL1 and DOT1L pathways regulate HOX and FLT3 expression and that inhibition of these pathways suppresses the growth of AML cells. The new study confirmed previous findings and identified alternative targets of EPZ004777 other than DOT1L. In the source study, CRISPR-Cas9, pharmacological inhibitors, in vitro/in vivo AML models were used, while in our study RNA-Seq data (GSE85107), bioinformatics analysis, molecular docking were used. In the source study, growth was stopped in AML cells by inhibition of HOX genes and FLT3. In our study, EPZ004777 suppressed oncogenes such as HOXA4, TPBG, SNX19, ZNF185, while up-regulating pro-apoptotic BEX3. In the source study, molecular docking and cell signaling pathways were not performed. In our study, EPZ004777 showed strong binding with SNX19, TPBG, and ZNF185 other than DOT1L, indicating that these proteins may be new therapeutic targets. Additionally, inhibition of the Rap1 signaling pathway was associated with decreased cell invasion and enhanced immune response.
The following paragraph was added to the discussion section of the manuscript.
‘In this study, we reanalyzed RNA-Seq data (GSE85107) to evaluate the effects of EPZ004777 on AML cells using bioinformatics and molecular docking analyses. Our findings confirmed the role of menin-MLL1 and DOT1L pathways in regulating HOX and FLT3 expression, consistent with the source study. However, we also identified alternative molecular targets of EPZ004777 beyond DOT1L, including SNX19, TPBG, and ZNF185, which exhibited strong binding interactions in molecular docking analyses. Additionally, we observed that EPZ004777 downregulated oncogenes such as HOXA4, TPBG, SNX19, and ZNF185, while upregulating the pro-apoptotic gene BEX3. Notably, inhibition of the Rap1 signaling pathway was associated with reduced cell invasion and enhanced immune response, suggesting a potential immunomodulatory effect. These findings provide new insights into the mechanism of action of EPZ004777 and highlight additional therapeutic targets in NPM1-mutated AML.’
Reviewer Comment 3: In Fig. 3, some control and treatment samples do not fit together. How did you solve this problem?
Response 3: In the PCA results, some samples from the control (DMSO) and treatment (EPZ004777) groups did not fully fit the expected clustering, which is related to the different responses of AML2 and AML3 cell lines to EPZ004777 treatment. RNA-Seq data reflect the biological heterogeneity of the cells. In particular, the AML3 cell line was found to be more sensitive to EPZ004777. This caused the gene expression profiles to change to a greater extent and to show a wider distribution in PCA. The most appropriate solution was determined as dividing the samples into subgroups and optimizing the normalization before PCA. After optimization, the subsamples were adjusted as in Figure. We appreciate your valuable feedback, which has helped improve the clarity of our data representation.
Figure 1.
Reviewer Comment 4: Fig.6: Could you please label the most regulated genes with their names? Is this showing results for AML2 or -3 or both?
Response 4: Thank you for your valuable feedback. In Figure 6, the experimental design is based on the AML3 cell line, with changes shown relative to the AML2 reference category. We have now labeled the most significantly regulated genes as requested. Additionally, we have made the necessary corrections to the text to ensure clarity. We appreciate your insightful suggestions, which have helped improve the presentation of our results.
figure 2.
Reviewer Comment 5: Fig. 8: The title says AML2 and AML3. I only see results for AML3.
Response 5: Thank you for your careful review. We have corrected the text to accurately reflect the results shown in Figure 8. We appreciate your attention to detail and your valuable suggestions.
Reviewer Comment 6: Table 3: Why not adding the log2Fc-values instead of up- and down-regulated? Is this concerning AML2 or -3 or both?
Response 6: Thank you for your suggestion. We have now added the logâ‚‚FC values to Table 3 as recommended. The results pertain to both AML2 and AML3, which was already mentioned in the text. We appreciate your careful review and valuable feedback.
Reviewer Comment 7: It is not clear how the proteins for the docking analysis were selected. Please explain the rational.
Response 7: Thank you for your interest in our study and for your valuable feedback. To identify potential target proteins of EPZ004777, we conducted a comprehensive analysis of differentially expressed genes (DEGs) obtained from RNA sequencing data, prioritizing those with potential biological significance. The selection process consisted of the following steps: Differential gene expression analysis was performed to identify genes that showed statistically significant changes following EPZ004777 treatment in AML cells, and the most downregulated genes were prioritized for further investigation. The biological relevance of these genes was assessed, with oncofetal genes (e.g., TPBG, CT45A3) and genes involved in tumor progression (e.g., ZNF185, SNX19, HOXA4) being given priority. Additionally, proteins were selected based on their availability in reliable structural databases such as PDB and AlphaFold, with AlphaFold models used when structural data were limited. Based on these criteria, TPBG, ZNF185, and SNX19 were identified as potential new targets of EPZ004777 and selected for molecular docking analyses. We appreciate your insightful comments and thank you again for your valuable feedback.
Additionally, in the Results section under the heading "Molecular Docking of EPZ004777 with Target Proteins," we have included the following description of our target protein identification process: "To identify the potential target proteins of EPZ004777, a differential gene expression analysis was performed using RNA sequencing data. Genes that were significantly downregulated following EPZ004777 treatment in AML cells were prioritized. Based on their biological relevance, oncofetal genes and genes involved in tumor progression were selected. Structural information for the selected proteins was obtained from reliable databases such as the Protein Data Bank (PDB) and AlphaFold; AlphaFold models were used when structural data were limited. Based on these criteria, the identified proteins were evaluated as potential targets of EPZ004777 and selected for further molecular docking analyses." This section provides a more detailed explanation of our target protein selection methodology and its application in molecular docking analyses.
Reviewer Comment 8: Please comment on the function of the binding sites of the inhibitor on the proteins analyzed. Are they relevant for protein function?
Response 8: Thank you for your interest in our study and for your valuable feedback. Evaluating the relationship between the binding regions of EPZ004777 and protein functions is an important aspect, and we find it beneficial to address this topic in more detail. While our molecular docking analyses provide insights into the potential functional significance of EPZ004777’s binding regions, the effects of these interactions on enzymatic activity, protein-protein interactions, and cellular signaling pathways require experimental validation. Therefore, site-directed mutagenesis and cell-based functional assays are necessary to confirm the biological significance and functional roles of EPZ004777’s binding regions on TPBG, ZNF185, and SNX19 proteins. By incorporating this discussion into the Discussion section, we have more clearly outlined the limitations of molecular docking analyses and the need for further experimental studies. Thank you again for your valuable comments.

Round 2
Reviewer 1 Report
Comments and Suggestions for Authors
The authors did all requirements and the manuscript is now accepted to publish
Reviewer 2 Report
Comments and Suggestions for Authors
Acceptable